# Uniform-PAC Bounds for Reinforcement Learning with Linear Function Approximation

**Jiafan He**
Department of Computer Science
University of California, Los Angeles
CA 90095, USA
jiafanhe19@ucla.edu

**Dongruo Zhou**
Department of Computer Science
University of California, Los Angeles
CA 90095, USA
drzhou@cs.ucla.edu

**Quanquan Gu**
Department of Computer Science
University of California, Los Angeles
CA 90095, USA
qgu@cs.ucla.edu

## Abstract

We study reinforcement learning (RL) with linear function approximation. Existing algorithms for this problem only have high-probability regret and/or Probably Approximately Correct (PAC) sample complexity guarantees, which cannot guarantee the convergence to the optimal policy. In this paper, in order to overcome the limitation of existing algorithms, we propose a new algorithm called FLUTE, which enjoys uniform-PAC convergence to the optimal policy with high probability. The uniform-PAC guarantee is the strongest possible guarantee for reinforcement learning in the literature, which can directly imply both PAC and high probability regret bounds, making our algorithm superior to all existing algorithms with linear function approximation. At the core of our algorithm is a novel minimax value function estimator and a multi-level partition scheme to select the training samples from historical observations. Both of these techniques are new and of independent interest.

## 1  Introduction

Designing efficient reinforcement learning (RL) algorithms for environments with large state and action spaces is one of the main tasks in the RL community. To achieve this goal, function approximation, which uses a class of predefined functions to approximate either the value function or transition dynamic, has been widely studied in recent years. Specifically, a series of recent works [11, 13, 18, 24, 3, 27] have studied RL with linear function approximation with provable guarantees. They show that with linear function approximation, one can either obtain a sublinear regret bound against the optimal value function [13, 24, 3, 27] or a polynomial sample complexity bound [14] (Probably Approximately Correct (PAC) bound for short) in finding a near-optimal policy [11, 18].

However, neither the regret bound or PAC bound is a perfect performance measure. As discussed in detail by [7], these two measures fail to guarantee the convergence to the optimal policy. Therefore, an algorithm with high probability regret and/or PAC bound guarantees do not necessarily learn the optimal policy, and can perform badly in practice. In detail, one can face the following two situations:

- An algorithm with a sublinear regret suggests that the summation of the suboptimality gaps $\Delta_t$ (the gap between the values of the current adapted policy and optimal policy, see Definition 3.3

35th Conference on Neural Information Processing Systems (NeurIPS 2021).

for details.) in the first $T$ rounds is bounded by $o(T)$. However, this algorithm may be arbitrarily suboptimal infinitely times [1], thus it fails to converge to the optimal policy.

- An algorithm is $(\epsilon, \delta)$-PAC suggests that with probability at least $1 - \delta$, the number of suboptimality gaps $\Delta_t$ that are greater than $\epsilon$ will be at most polynomial in $\epsilon$ and $\log(1/\delta)$. The formal definition of $(\epsilon, \delta)$-PAC can be found in Definition 3.4. However, this algorithm may still have gaps satisfying $\epsilon/2 < \Delta_t < \epsilon$ infinitely often, thus fails to converge to the optimal policy.

To overcome the limitations of regret and PAC guarantees, Dann et al. [7] proposed a new performance measure called uniform-PAC, which is a strengthened notion of the PAC framework. Specifically, an algorithm is uniform-PAC if there exists a function of the target accuracy $\epsilon$ and the confidence parameter $\delta$ that upper bounds the number of suboptimality gaps satisfying $\Delta_t > \epsilon$ *simultaneously* for all $\epsilon > 0$ with probability at least $1 - \delta$. The formal definition of uniform-PAC can be found in Definition 3.6. Algorithms that are uniform-PAC converge to an optimal policy with high probability, and yield both PAC and high probability regret bounds. In addition, they proposed a UBEV algorithm for learning tabular MDPs, which is uniform-PAC. Nevertheless, UBEV is designed for tabular MDPs, and it is not clear how to incorporate function approximation into UBEV to scale it up for large (or even infinite) state and action space. Therefore, a natural question arises:

*Can we design a provable efficient uniform-PAC RL algorithm with linear function approximation?*

In this work, we answer this question affirmatively. In detail, we propose new algorithms for both contextual linear bandits and linear Markov decision processes (MDPs) [22, 13]. Both of them are uniform-PAC, and their sample complexity is comparable to that of the state-of-the-art algorithms which are not uniform-PAC. Our key contributions are highlighted as follows.

- We begin with contextual linear bandits problem as a "warm-up" example of the RL with linear function approximation (with horizon length equals 1). We propose a new algorithm called uniform-PAC OFUL (UPAC-OFUL), and show that our algorithm is uniform-PAC with $\widetilde{O}(d^2/\epsilon^2)$ sample complexity, where $d$ is the dimension of contexts and $\epsilon$ is the accuracy parameter. In addition, this result also implies an $\widetilde{O}(d\sqrt{T})$ regret in the first $T$ round and matches the result of OFUL algorithm [1] up to a logarithmic factor. The key idea of our algorithm is a novel minimax linear predictor and a multi-level partition scheme to select the training samples from past observations. To the best of our knowledge, this is the first algorithm with a uniform PAC-bound for contextual bandits problems.

- We also consider RL with linear function approximation in episodic linear MDPs, where the transition kernel admits a low-rank factorization. We propose an algorithm dubbed uniForm-PAC Least-sqUare value iTEration (FLUTE), which adapts the novel techniques we developed in the contextual linear bandits setting, and show that our algorithm is uniform-PAC with $\widetilde{O}(d^3 H^5/\epsilon^2)$ sample complexity, where $d$ is the dimension of the feature mapping, $H$ is the length of episodes and $\epsilon$ is the accuracy parameter. This result further implies an $\widetilde{O}(\sqrt{d^3 H^4 T})$ regret in the first $T$ steps and matches the result of LSVI-UCB algorithm [13] up to a $\sqrt{H}$-factor, while LSVI-UCB is not uniform-PAC. Again, FLUTE is the first uniform-PAC RL algorithm with linear function approximation.

**Notation** We use lower case letters to denote scalars, and use lower and upper case bold face letters to denote vectors and matrices respectively. For any positive integer $n$, we denote by $[n]$ the set $\{1, \ldots, n\}$. For a vector $\mathbf{x} \in \mathbb{R}^d$, we denote by $\|\mathbf{x}\|_1$ the Manhattan norm and denote by $\|\mathbf{x}\|_2$ the Euclidean norm. For a vector $\mathbf{x} \in \mathbb{R}^d$ and matrix $\mathbf{\Sigma} \in \mathbb{R}^{d \times d}$, we define $\|\mathbf{x}\|_{\mathbf{\Sigma}} = \sqrt{\mathbf{x}^\top \mathbf{\Sigma} \mathbf{x}}$. For two sequences $\{a_n\}$ and $\{b_n\}$, we write $a_n = O(b_n)$ if there exists an absolute constant $C$ such that $a_n \leq C b_n$. We use $\widetilde{O}(\cdot)$ to further hide the logarithmic factors. For logarithmic regret, we use $\widetilde{O}(\cdot)$ to hide all logarithmic terms except $\log T$.

---

[1]Suppose the suboptimality gaps satisfy $\Delta_t = \mathbb{1}\{t = i^2, i = 1, \ldots\}$, then the regret in the first $T$ rounds is upper bounded by $O(\sqrt{T})$, and the constant 1-gap will appear infinitely often

## 2 Related Work

### 2.1 Linear bandits

There is a series of works focusing on the stochastic linear bandits problem. These works can be categorized into two groups: the works aim at providing sublinear regret guarantee and the works providing PAC bound for linear best-arm identification problem. In detail, for finite action set with $K$ arms, Auer [2] proposed a SupLinRel algorithm which achieves an $O(\sqrt{dT\log^3(TK)})$ regret, where $K$ is the number of arms. Chu et al. [5] proposed a SupLinUCB algorithm which has the same regret bound as SupLinRel, but is easier to implement. Li et al. [17] proposed a Variable-Confidence-Level (VCL) SupLinUCB algorithm and improved the regret bound to $O(\sqrt{dT\log T\log K})$. For infinite action set, Dani et al. [6] proposed a Confidence Ball algorithm with an $O(d\sqrt{T\log^3 T})$ regret and proved an $\Omega(d\sqrt{T})$ lower bound. Abbasi-Yadkori et al. [1] proposed OFUL algorithm and improved the regret bound to $O(d\sqrt{T\log^2 T})$. When the reward has a bounded variance, Zhou et al. [27], Zhang et al. [25] proposed algorithms with variance-aware confident sets, and obtained tight variance-dependent regret bounds. For the best-arm identification problem, to find an arm which is $\epsilon$-suboptimal, Soare et al. [19] proposed a G-allocation strategy with an $\widetilde{O}(d/\epsilon^2)$ sample complexity. Karnin [15] proposed an Explore-Verify framework which improves the sample complexity by some logarithmic factors. Xu et al. [21] proposed a LinGapE algorithm whose sample complexity matches the lower bound up to some $K$ factors. Tao et al. [20] proposed an ALBA algorithm which improves the sample complexity to have a linear dimension dependence. Fiez et al. [8] studied transductive linear bandits and proposed an algorithm with sample complexity similar to linear bandits. Compared with best-arm identification, the contextual bandits setting we focus on is more challenging since the action set will change at each round.

### 2.2 RL with linear function approximation

Recently, a line of work focuses on analyzing RL with linear function approximation. To mention a few, Jiang et al. [11] studied MDPs with low Bellman rank and proposed an OLIVE algorithm, which has the PAC guarantee. Yang and Wang [22] studied the linear transition model and proposed a sample-optimal Q-learning method with a generative model. Jin et al. [13] studied the linear MDP model and proposed an LSVI-UCB algorithm under the online RL setting (without a generative model) with $\widetilde{O}(\sqrt{d^3H^3T})$ regret. Later, Zanette and Brunskill [23] studied the low inherent Bellman error model and proposed an ELEANOR algorithm with a better regret $\widetilde{O}(dH\sqrt{T})$ using a global planning oracle. Modi et al. [18] studied the linearly combined model ensemble and proposed a provable sample-efficient algorithm. Jia et al. [10], Ayoub et al. [3] studied the linear mixture MDPs and proposed a UCRL-VTR algorithm with an $\widetilde{O}(d\sqrt{H^3T})$ regret. Recently Zhou et al. [27] improved the regret bound to $\widetilde{O}(dH\sqrt{T})$ with a new algorithm design and a new Bernstein inequality. However, all of these works aim at deriving PAC sample complexity guarantee or regret bound, and none of them has the uniform PAC guarantee for learning MDPs with linear function approximation. Our work will fill this gap in the linear MDP setting [22, 13].

## 3 Preliminaries

We consider episodic Markov Decision Processes (MDPs) in this work. Each episodic MDP is denoted by a tuple $M(\mathcal{S}, \mathcal{A}, H, \{r_h\}_{h=1}^H, \{\mathbb{P}_h\}_{h=1}^H)$. Here, $\mathcal{S}$ is the state space, $\mathcal{A}$ is the finite action space, $H$ is the length of each episode, $r_h : \mathcal{S} \times \mathcal{A} \to [0, 1]$ is the reward function at stage $h$ and $\mathbb{P}_h(s'|s, a)$ is the transition probability function at stage $h$ which denotes the probability for state $s$ to transfer to state $s'$ with action $a$ at stage $h$. A policy $\pi : \mathcal{S} \times [H] \to \mathcal{A}$ is a function which maps a state $s$ and the stage number $h$ to an action $a$. For any policy $\pi$ and stage $h \in [H]$, we define the action-value function $Q_h^\pi(s, a)$ and value function $V_h^\pi(s)$ as follows

$$Q_h^\pi(s, a) = r_h(s, a) + \mathbb{E}\bigg[\sum_{h'=h+1}^H r_{h'}\big(s_{h'}, \pi(s_{h'}, h')\big)\big|s_h = s, a_h = a\bigg], \ V_h^\pi(s) = Q_h^\pi(s, \pi(s, h)),$$

where $s_{h'+1} \sim \mathbb{P}_h(\cdot|s_{h'}, a_{h'})$. We define the optimal value function $V_h^*$ and the optimal action-value function $Q_h^*$ as $V_h^*(s) = \max_\pi V_h^\pi(s)$ and $Q_h^*(s, a) = \max_\pi Q_h^\pi(s, a)$. By definition, the

value function $V_h^\pi(s)$ and action-value function $Q_h^\pi(s,a)$ are bounded in $[0, H]$. For any function $V : \mathcal{S} \to \mathbb{R}$, we denote $[\mathbb{P}_h V](s,a) = \mathbb{E}_{s' \sim \mathbb{P}_h(\cdot|s,a)} V(s')$. Therefore, for each stage $h \in [H]$ and policy $\pi$, we have the following Bellman equation, as well as the Bellman optimality equation:

$$Q_h^\pi(s,a) = r_h(s,a) + [\mathbb{P}_h V_{h+1}^\pi](s,a), \; Q_h^*(s,a) = r_h(s,a) + [\mathbb{P}_h V_{h+1}^*](s,a), \qquad (3.1)$$

where $V_{H+1}^\pi = V_{H+1}^* = 0$. At the beginning of the episode $k$, the agent determines a policy $\pi_k$ to be followed in this episode. At each stage $h \in [H]$, the agent observes the state $s_h^k$, chooses an action following the policy $\pi_k$ and observes the next state with $s_{h+1}^k \sim \mathbb{P}_h(\cdot|s_h^k, a_h^k)$.

We consider linear function approximation in this work. Therefore, we make the following linear MDP assumption, which is firstly proposed in [22, 13].

**Assumption 3.1.** MDP $\mathcal{M}(\mathcal{S}, \mathcal{A}, H, \{r_h\}_{h=1}^H, \{\mathbb{P}_h\}_{h=1}^H)$ is a linear MDP such that for any stage $h \in [H]$, there exists an unknown vector $\boldsymbol{\mu}_h$, an unknown measure $\boldsymbol{\theta}_h(\cdot) : \mathcal{S} \to \mathbb{R}^d$ and a known feature mapping $\boldsymbol{\phi} : \mathcal{S} \times \mathcal{A} \to \mathbb{R}^d$, such that for each $(s,a) \in \mathcal{S} \times \mathcal{A}$ and $s' \in \mathcal{S}$,

$$\mathbb{P}_h(s'|s,a) = \langle \boldsymbol{\phi}(s,a), \boldsymbol{\theta}_h(s') \rangle, r_h(s,a) = \langle \boldsymbol{\phi}(s,a), \boldsymbol{\mu}_h \rangle.$$

For simplicity, we assume that $\boldsymbol{\mu}_h$, $\boldsymbol{\theta}_n(\cdot)$ and $\boldsymbol{\phi}(\cdot, \cdot)$ satisfy $\|\boldsymbol{\phi}(s,a)\|_2 \le 1$ for all $s, a$, $\|\boldsymbol{\mu}_h\|_2 \le \sqrt{d}$ and $\|\boldsymbol{\theta}_h(\mathcal{S})\|_2 \le \sqrt{d}$. The linear MDP assumption automatically suggests that for any policy $\pi$, the action-value function $Q_h^\pi$ is always a linear function of the given feature mapping $\boldsymbol{\phi}$, which is summarized in the following proposition.

**Proposition 3.2** (Proposition 2.3, [13]). For any policy $\pi$, there exist weights $\{\mathbf{w}_h^\pi\}_{h=1}^H$ such that for any $s, a, h \in \mathcal{S} \times \mathcal{A} \times [H]$, $Q_h^\pi(s,a) = \langle \boldsymbol{\phi}(s,a), \mathbf{w}_h^\pi \rangle$.

Next we define the regret and $(\epsilon, \delta)$-PAC formally.

**Definition 3.3.** For an RL algorithm `Alg`, we define its regret on learning an MDP $M(\mathcal{S}, \mathcal{A}, H, r, \mathbb{P})$ in the first $K$ episodes as the sum of the suboptimality for episode $k = 1, \dots, K$,

$$\text{Regret}(K) = \sum_{k=1}^K V_1^*(s_1^k) - V_1^{\pi_k}(s_1^k),$$

where $\pi_k$ is the policy in the $k$-th episode .

**Definition 3.4.** For an RL algorithm `Alg` and a fixed $\epsilon$, let $\pi_1, \pi_2, \dots$ be the policies generated by `Alg`. Let $N_\epsilon = \sum_{k=1}^\infty \mathbb{1}\{V_1^*(s_1^k) - V_1^{\pi_k}(s_1^k) > \epsilon\}$ be the number of episodes whose suboptimality gap is greater than $\epsilon$. Then we say `Alg` is $(\epsilon, \delta)$-PAC with sample complexity $f(\epsilon, \delta)$ if

$$\mathbb{P}(N_\epsilon > f(\epsilon, \delta)) \le \delta.$$

**Remark 3.5.** Dann et al. [7] suggested that an algorithm with a sublinear regret is not necessarily to be an $(\epsilon, \delta)$-PAC algorithm. However, with some modification, Jin et al. [12] showed that any algorithm with a sublinear regret can be converted to a *new* algorithm which is $(\epsilon, \delta)$-PAC, which does not contradict with the claim by [7]. For example, Ghavamzadeh et al. [9] and Zhang et al. [26] proposed algorithms with $\widetilde{O}(\sqrt{SAHT})$ regret, and both algorithms can be converted into new algorithms which are $(\epsilon, \delta)$-PAC with sample complexity $\widetilde{O}(SAH^2/\epsilon^2)$.

Both regret and PAC guarantees are not perfect. As Dann et al. [7] showed, an algorithm with sub-linear regret or $(\delta, \epsilon)$-PAC bound may fail to converge to the optimal policy. For an $(\delta, \epsilon)$-PAC algorithm with $\Delta_t = \epsilon/2 (t \in \mathbb{N})$, it still has linear regret $O(\epsilon T)$ and will never converge to the optimal policy. For an algorithm with a sub-linear regret bound, a constant sub-optimality gap may still occur infinite times. Therefore, Dann et al. [7] proposed uniform-PAC algorithms, which are defined formally as follows.

**Definition 3.6.** For an RL algorithm `Alg`, let $\pi_1, \pi_2, \dots$ be the policies generated by `Alg`. Let $N_\epsilon = \sum_{k=1}^\infty \mathbb{1}\{V_1^*(s_1^k) - V_1^{\pi_k}(s_1^k) > \epsilon\}$ be the number of episodes whose suboptimality gap is greater than $\epsilon$. We say `Alg` is uniform-PAC for some $\delta \in (0,1)$ with sample complexity $f(\epsilon, \delta)$ if

$$\mathbb{P}(\exists \epsilon > 0, \; N_\epsilon > f(\epsilon, \delta)) \le \delta.$$

The following theorem suggests that a uniform-PAC algorithm is automatically a PAC algorithm and an algorithm with sublinear regret.

---
**Algorithm 1** Uniform-PAC OFUL (UPAC-OFUL)
---
**Require:** Regularization parameter $\lambda$, confidence radius $\beta_l(l \in \mathbb{N})$
1: Set $\mathcal{C}^l \leftarrow \emptyset, l \in \mathbb{N}$ and the total level $S_1 = 1$
2: **for** round $k = 1, 2, ..$ **do**
3:     **for** all level $l \in [S_k]$ **do**
4:         Set $\boldsymbol{\Sigma}_k^l = \lambda \mathbf{I} + \sum_{i \in \mathcal{C}^l} \mathbf{x}_i \mathbf{x}_i^\top$
5:         Set $\mathbf{b}_k^l = \sum_{i \in \mathcal{C}^l} \mathbf{x}_i r_i$ and $\mathbf{w}_k^l = (\boldsymbol{\Sigma}_k^l)^{-1} \mathbf{b}_k^l$
6:     **end for**
7:     Receive the action set $\mathcal{D}_k$
8:     Choose action $\mathbf{x}_k \leftarrow \operatorname{argmax}_{\mathbf{x} \in \mathcal{D}_k} \min_{1 \le l \le S_k} (\mathbf{w}_k^l)^\top \mathbf{x} + \beta_l \sqrt{\mathbf{x}^\top (\boldsymbol{\Sigma}_k^l)^{-1} \mathbf{x}}$
9:     Set level $l_k = 1$
10:    **while** $\sqrt{\mathbf{x}_k^\top (\boldsymbol{\Sigma}_k^{l_k})^{-1} \mathbf{x}_k} \le 2^{-l_k}$ and $l_k \le S_k$ **do**
11:       $l_k \leftarrow l_k + 1$
12:    **end while**
13:    Add the new element $k$ to the set $\mathcal{C}^{l_k}$ and receive the reward $r_k$
14:    Set the total level $S_{k+1}$ as $S_{k+1} = \max_{l:|\mathcal{C}^l|>0} l$
15: **end for**
---

**Theorem 3.7** (Theorem 3, [7]). If an algorithm Alg is uniform-PAC for some $\delta \ge 0$, with sample complexity $\widetilde{O}(C_1/\epsilon + C_2/\epsilon^2)$, where $C_1, C_2$ are constant and depend only on $S, A, H, \log(1/\delta)$. Then, we have the following results:

- 1: Alg will converge to optimal policies with high probability at least $1-\delta$: $\mathbb{P}\big(\lim_{k \to +\infty} V_1^*(s_1^k) - V_1^{\pi_k}(s_1^k) = 0\big) \ge 1 - \delta$

- 2: With probability at least $1 - \delta$, for each $K \in \mathbb{N}$, the regret for Alg in the first $K$ episodes is upper bounded by $\widetilde{O}(\sqrt{C_2 K} + C_1 + C_2)$.

- 3: For each $\epsilon \ge 0$, Alg is also $(\epsilon, \delta)$-PAC with the same sample complexity $\widetilde{O}(C_1/\epsilon + C_2/\epsilon^2)$.

Theorem 3.7 suggests that uniform-PAC is stronger than both the PAC and regret guarantees. In the remainder of this paper, we aim at developing uniform-PAC RL algorithms with linear function approximation.

## 4   Warm up: Uniform-PAC Bounds for Linear Bandits

To better illustrate the idea of our algorithm, in this section, we consider a contextual linear bandits problem, which can be regarded as a special linear MDP with $H = 1$. Let $\{\mathcal{D}_k\}_{k=1}^\infty$ be a fixed sequence of decision/action sets. At round $k$, the agent selects an action $\mathbf{x}_k \in \mathcal{D}_k$ by the algorithm $\mathcal{H}$ and then observes the reward $r_k = \langle \boldsymbol{\mu}^*, \mathbf{x}_k \rangle + \epsilon_k$, where $\boldsymbol{\mu}^* \in \mathbb{R}^d$ is a vector unknown to the agent and $\epsilon_k$ is a sub-Gaussian random noise. $\mathbf{x}_k, \epsilon_k, \boldsymbol{\mu}^*$ satisfy the following properties:

$$\forall k \in \mathbb{N}, \lambda \in \mathbb{R}, \ \mathbb{E}\big[e^{\lambda \epsilon_k} | \mathbf{x}_{1:k}, \epsilon_{1:k-1}\big] \le \exp(\lambda^2/2), \|\mathbf{x}_k\|_2 \le 1, \|\boldsymbol{\mu}^*\|_2 \le 1. \tag{4.1}$$

Our goal is to design an $(\epsilon, \delta)$-uniform-PAC algorithm with sample complexity $f(\epsilon, \delta)$ such that

$$\mathbb{P}\bigg(\exists \epsilon > 0, \ \sum_{k=1}^\infty \mathbb{1}\left\{\Delta_k := \max_{\mathbf{x} \in \mathcal{D}_k} \langle \boldsymbol{\mu}^*, \mathbf{x} \rangle - \langle \boldsymbol{\mu}^*, \mathbf{x}_k \rangle > \epsilon\right\} > f(\epsilon, \delta)\bigg) < \delta,$$

where $\Delta_k := \max_{\mathbf{x} \in \mathcal{D}_k} \langle \boldsymbol{\mu}^*, \mathbf{x} \rangle - \langle \boldsymbol{\mu}^*, \mathbf{x}_k \rangle$ denotes the suboptimality at round $k$.

Here we assume the weight vector $\boldsymbol{\mu}^*$ satisfies $\|\boldsymbol{\mu}^*\|_2 \le 1$, to be consistent with the assumption made in Abbasi-Yadkori et al. [1]. Our assumption can be easily relaxed to the general $\|\boldsymbol{\mu}^*\|_2 \le B$ case with an additional $\log B$ factor in the sample complexity, as can be seen in the following analysis.

**Why existing algorithms fail to be uniform-PAC?** Before proposing our algorithm, it is natural to ask whether existing methods have already been uniform-PAC. We take OFUL [1] for example,

which is the state-of-the-art linear bandit algorithm in our setting. At round $k$, OFUL constructs an optimistic estimation of the true linear function $\langle \boldsymbol{\mu}^*, \mathbf{x} \rangle$, by doing linear regression over all past $k$ selected actions $\mathbf{x}_i, 1 \leq i \leq k$ and their corresponding rewards. The optimistic estimation has a closed-form as the summation of the linear regression predictor and a quadratic confidence bound $\mathbf{w}_k^\top \mathbf{x} + \alpha \sqrt{\mathbf{x}^\top \boldsymbol{\Sigma}_k^{-1} \mathbf{x}}$, where $\boldsymbol{\Sigma}_k = \lambda \mathbf{I} + \sum_{i=1}^{k-1} \mathbf{x}_i \mathbf{x}_i^\top$ [16]. Following the standard analysis of OFUL in [1], we obtain the following upper confidence bound of the suboptimality gap $\Delta_k$:

$$\text{With probability at least } 1 - \delta, \ \forall k > 0, \ \Delta_k = O\big(\sqrt{d \log(k/\delta)} \|\mathbf{x}_k\|_{\boldsymbol{\Sigma}_k^{-1}}\big), \tag{4.2}$$

where the $\log k$ is due to the fact that OFUL makes use of *all* past $k$ observed actions. Since the agent can only say whether an arm is good or not based on the confidence bound of $\Delta_k$, due to the existence of the $\log k$ term in (4.2), the bounds on the suboptimality gap for the "good" arms may be large (since $\log k$ grows as $k$ increases). That makes the agent fail to recognize those "good" arms and instead pull the "bad" arms infinite times, which suggests that OFUL is not a uniform-PAC algorithm. For other algorithms, they either need to know the total round $T$ before running the algorithm [5], or need to assume that the decision sets $\mathcal{D}_k$ are identical (e.g., algorithms for best-arm identification [19]), thus none of them fits into our setting.

**Key techniques of our algorithm.** In order to address the aforementioned issue, we proposed UPAC-OFUL in Algorithm 1. The key idea of Algorithm 1 is to divide all the historical observed data into non-overlapping sets $\mathcal{C}^l$, while each $\mathcal{C}^l$ only includes *finite* past historical observed actions. This helps successfully avoid the $\log k$ term appearing in the confidence bound in [1]. Then at round $k$, Algorithm 1 only constructs optimistic estimation of $\langle \boldsymbol{\mu}^*, \mathbf{x}_k \rangle$ over the first $S_k$ sets $\mathcal{C}^l$ individually, where $S_k$ is the number of non-empty sets $\mathcal{C}^l$. In detail, the optimistic estimation over $\mathcal{C}^l$ has the form

$$(\mathbf{w}_k^l)^\top \mathbf{x} + \beta_l \sqrt{\mathbf{x}^\top (\boldsymbol{\Sigma}_k^l)^{-1} \mathbf{x}}, \tag{4.3}$$

where $\boldsymbol{\Sigma}_k^l$ is the covariance matrix for actions in set $\mathcal{C}^l$ (Line 4), and $\mathbf{w}_k^l$ is the estimation of $\boldsymbol{\mu}^*$ obtained by ridge regression defined in Line 5. Meanwhile, for a newly selected action $\mathbf{x}_k$, Algorithm 1 needs to decide which $\mathcal{C}^l$ it should be added to. Inspired by [5], Algorithm 1 tests the "uncertainty" of $\mathbf{x}_k$ against $\mathcal{C}^l$, by calculating its confidence bound $\|\mathbf{x}_k\|_{(\boldsymbol{\Sigma}_k^l)^{-1}}$. Then Algorithm 1 adds $\mathbf{x}_k$ to the *lowest possible* level where the "uncertainty" is larger than a certain threshold (i.e., $\|\mathbf{x}_k\|_{(\boldsymbol{\Sigma}_k^l)^{-1}} > 2^{-l}$). Such a selection rule guarantees two things simultaneously. First, it ensures that the cardinality of each $\mathcal{C}^l$ is finite, due to the fact that the summation of $\|\mathbf{x}_k\|_{(\boldsymbol{\Sigma}_k^l)^{-1}}$ can be properly bounded. Second, it also guarantees that the "uncertainty" of the reward corresponding to $\mathbf{x}_k$ is still small, since by the level selection rule we have $\|\mathbf{x}_k\|_{(\boldsymbol{\Sigma}_k^{l-1})^{-1}} \leq 2^{-(l-1)}$. Lastly, to make use of all $S_k$ optimistic estimations, Algorithm 1 constructs the final predictor as the minimal value of $S_k$ individual predictor (4.3) over $\mathcal{C}^l$ (Line 8). Since each individual predictor is a valid upper bound of the true function, the minimum of them is still valid and tighter than each of them (except the smallest one), which makes it possible to provide a stronger uniform-PAC guarantee.

The following theorem shows that Algorithm 1 is indeed uniform-PAC.

**Theorem 4.1.** For any $\delta \in (0, 1)$, if we set $\lambda = 1$ and $\beta_l = 6\sqrt{dl \log(dl/\delta)}$ for every level $l \in \mathbb{N}$, then there exists a constant $C$ such that with probability at least $1 - \delta$, for all $\epsilon > 0$, the number of rounds in Algorithm 1 which have sub-optimality no less than $\epsilon$ is bounded by

$$\sum_{k=1}^\infty \mathbb{1}\left\{ \max_{\mathbf{x} \in \mathcal{D}_k} \langle \boldsymbol{\mu}^*, \mathbf{x} \rangle - \langle \boldsymbol{\mu}^*, \mathbf{x}_k \rangle > \epsilon \right\} \leq \frac{C d^2 \log^3\big(d/(\delta\epsilon)\big)}{\epsilon^2}.$$

**Remark 4.2.** Theorem 4.1 suggests that Algorithm 1 is uniform-PAC with sample complexity $O(d^2 \log^3\big(d/(\delta\epsilon)\big)/\epsilon^2)$. According to Theorem 3.7, this new algorithm will converge to the optimal policy. Theorem 4.1 also implies an $\widetilde{O}(d\sqrt{T})$ regret for infinite-arm linear bandit problem. This result matches the lower bound $\Omega(d\sqrt{T})$ [6] up to a logarithmic factor. Furthermore, Theorem 4.1 implies that Algorithm 1 is an $(\epsilon, \delta)$-PAC algorithm with sample complexity $\widetilde{O}(d^2/\epsilon^2)$. Specifically, if we set $\epsilon = \Delta_{\min}$, Theorem 4.1 implies an $\widetilde{O}(d^2/\Delta_{\min}^2)$ sample complexity to identify the best arm[2], which matches the sample complexity $\widetilde{O}(d \log K/\Delta_{\min}^2)$ in [19] when $K = \Theta(2^d)$.

---

[2]Soare et al. [19] denoted by $\Delta_{\min}$ the gap of the rewards between the best arm and the second-best arm. In this setting, the sample complexity to find the best arm is identical to the sample complexity to find an $\epsilon < \Delta_{\min}$ sub-optimal arm.

# 5  Uniform-PAC Bounds for Linear MDPs

In this section, we propose our new FLUTE algorithm (Algorithm 2) for learning linear MDPs, and provide its theoretical guarantee.

**Intuition behind FLUTE**  At a high level, FLUTE inherits the structure of Least-Square Value Iteration with UCB (LSVI-UCB) proposed in [13]. The Bellman optimality equation gives us the following equation:

$$r_h(s,a) + [\mathbb{P}_h V_{h+1}^*](s,a) = Q_h^*(s,a) = \langle \boldsymbol{\theta}_h^*, \boldsymbol{\phi}(s,a) \rangle, \tag{5.1}$$

where the second equality holds due to Proposition 3.2. (5.1) suggests that in order to learn $Q_h^*$, it suffices to learn $\boldsymbol{\theta}_h^*$, which can be roughly regarded as the unknown vector of a linear bandits problem with actions $\boldsymbol{\phi}(s,a)$ and rewards $r_h(s,a) + V_{h+1}^*(s')$, where $(s,a,s')$ belongs to some set $\mathcal{C}$. Since $V_{h+1}^*$ is unknown, we use its estimation $V_{h+1}$ to replace it. Therefore, we can apply Algorithm 1 to this equivalent linear bandits problem to obtain our uniform-PAC RL algorithm FLUTE.

**Details of FLUTE**  We now describe the details of FLUTE. For each stage $h$, FLUTE maintains non-overlapping index set $\{\mathcal{C}_h^l\}_l$, each $\mathcal{C}_h^l$ contains state-action-next-state triples $(s_h^i, a_h^i, s_{h+1}^i)$. Let $S_1 = 1$ and $S_k$ denote the number of non-empty sets $\{\mathcal{C}_1^l\}_l$ at episode $k$ for $k \geq 2$. Instead of maintaining only one estimated optimal value function $V_{k,h}$ and action-value function $Q_{k,h}$ [13], FLUTE maintains *a group* of estimated value functions $\{V_{k,h}^l\}_l$ and action-value functions $\{Q_{k,h}^l\}_l$. In detail, at stage $h$, given $\{V_{k,h+1}^l\}_l$, FLUTE calculates $\mathbf{w}_{k,h}^l$ as the minimizer of the ridge regression problem with training dataset $(s_h^i, a_h^i, s_{h+1}^i) \in \mathcal{C}_h^l$ and targets $V_{k,h+1}^l(s_{h+1}^i)$ (Line 8), and defines $Q_{k,h}^l$ as the summation of the linear predictor $(\mathbf{w}_{k,h}^l)^\top \boldsymbol{\phi}(s,a)$ and a quadratic confidence bonus $\beta_l \sqrt{\boldsymbol{\phi}(s,a)^\top (\boldsymbol{\Sigma}_{k,h}^l)^{-1} \boldsymbol{\phi}(s,a)}$ (Line 9), where $\beta_l$ is the confidence radius for level $l$ and $\boldsymbol{\Sigma}_{k,h}^l$ is the covariance matrix for contexts in set $\mathcal{C}_h^l$. Then FLUTE defines value function $V_{k,h}^l$ as the maximum of the minimal value over the first $l$ action-value functions (Line 12). The max-min structure is similar to its counterpart for linear bandits in Algorithm 1, which provides a tighter estimation of the optimal value function and is pivotal to achieve uniform-PAC guarantee.

After constructing action-value functions $\{Q_{k,h}^l\}_l$, FLUTE executes the greedy policy induced by the minimal of action-value function $Q_{k,h}^l$ over $1 \leq l \leq l_{h-1}^k - 1$, where $l_0^k = S_k + 1$ and $l_{h-1}^k$ is the level of the set $\mathcal{A}_{h-1}^l$ that we add the triple $(s_{h-1}^k, a_{h-1}^k, s_h^k)$. After obtaining $(s_h^k, a_h^k, s_{h+1}^k)$, to decide which set $\mathcal{C}_h^l$ should this triple be added, FLUTE calculates the confidence bonus $\sqrt{\boldsymbol{\phi}(s_h^k, a_h^k)^\top (\boldsymbol{\Sigma}_{k,h}^l)^{-1} \boldsymbol{\phi}(s_h^k, a_h^k)}$ and puts it into the $l$-th set if the confidence bonus is large (Line 19), similar to that of Algorithm 1.

The following theorem shows that FLUTE is uniform-PAC for learning linear MDPs.

**Theorem 5.1.** Under Assumption 3.1, there exists a positive constant $C$ such that for any $\delta \in (0,1)$, if we set $\lambda = 1$ and $\beta_l = CdHl\sqrt{\log(dlH/\delta)}$, then with probability at least $1 - \delta$, for all $\epsilon > 0$, we have

$$\sum_{k=1}^{\infty} \mathbb{1}\{V_1^*(s_1^k) - V_1^{\pi_k} \geq \epsilon\} = O(d^3 H^5 \log^4(dH/(\delta\epsilon))/\epsilon^2).$$

**Remark 5.2.** Theorem 5.1 suggests that algorithm FLUTE is uniform-PAC with sample complexity $O(d^3 H^5 \log^4(dH/(\delta\epsilon))/\epsilon^2)$. According to Theorem 3.7, FLUTE will converge to the optimal policy with high probability. Theorem 5.1 also implies an $\widetilde{O}(\sqrt{d^3 H^4 T})$ regret for linear MDPs. This result matches the regret bound $\widetilde{O}(\sqrt{d^3 H^3 T})$ of LSVI-UCB [13] up to a $\sqrt{H}$-factor. Furthermore, Theorem 5.1 also implies FLUTE is an $(\epsilon, \delta)$-PAC with sample complexity $\widetilde{O}(d^3 H^5/\epsilon^2)$, which matches the $O(d^3 H^3/\epsilon^2)$ sample complexity of LSVI-UCB up to an $H$ factor.

**Computational complexity**  As shown in Jin et al. [13], the time complexity of LSVI-UCB is $O(d^2 AHK^2)$. Compared with the LSVI-UCB algorithm, Algorithm 2 maintains non-overlapping index sets $\{\mathcal{C}_h^l\}$ and computes the corresponding optimistic value function for each level $\ell$. Without further assumption on the norm of $\boldsymbol{\phi}(s,a)$, the number of different levels in the first $K$ episodes is at

**Algorithm 2** Uniform PAC Least-Square Value Iteration (FLUTE)

**Require:** Regualarization parameter $\lambda$, confidence radius $\beta_l$
1: Set $\mathcal{C}_h^l \leftarrow \emptyset, l \in \mathbb{N}, h \in [H]$ and set the total level $S_1 = 1$
2: **for** episode $k = 1, 2, ..$ **do**
3:     Set $V_{k,H+1}^l(s, a) = 0$ for all state-action pair $(s, a) \in \mathcal{S} \times \mathcal{A}$ and all level $l \in [S_k]$
4:     **for** stage $h = H, H-1, .., 1$ **do**
5:         **for** all level $l \in [S_k]$ **do**
6:             Set $\mathbf{\Sigma}_{k,h}^l = \lambda \mathbf{I} + \sum_{i \in \mathcal{C}_h^l} \boldsymbol{\phi}(s_h^i, a_h^i) \boldsymbol{\phi}(s_h^i, a_h^i)^\top$
7:             Set $\mathbf{b}_{k,h}^l = \sum_{i \in \mathcal{C}_h^l} \boldsymbol{\phi}(s_h^i, a_h^i) \Big[ r_h(s_h^i, a_h^i) + V_{k,h+1}^l(s_{h+1}^i) \Big]$
8:             $\mathbf{w}_{k,h}^l \leftarrow (\mathbf{\Sigma}_{k,h}^l)^{-1} \mathbf{b}_{k,h}^l$
9:             $Q_{k,h}^l(s, a) \leftarrow \min \Big\{ H, (\mathbf{w}_{k,h}^l)^\top \boldsymbol{\phi}(s, a) + \beta_l \sqrt{\boldsymbol{\phi}(s, a)^\top (\mathbf{\Sigma}_{k,h}^l)^{-1} \boldsymbol{\phi}(s, a)} \Big\}$
10:         **end for**
11:         **for** all level $l \in [S_k]$ **do**
12:             $V_{k,h}^l(s) \leftarrow \max_a \min_{1 \le i \le l} Q_{k,h}^i(s, a)$
13:         **end for**
14:     **end for**
15:     Receive the initial state $s_1^k$ and set the current level $l_0^k = S_k + 1$
16:     **for** stage $h = 1, 2, .., H$ **do**
17:         Take action $a_h^k \leftarrow \mathrm{argmax}_a \min_{1 \le i \le l_{h-1}^k - 1} Q_{k,h}^i(s_h^k, a)$
18:         Set level $l_h^k = 1$
19:         **while** $\sqrt{\boldsymbol{\phi}(s_h^k, a_h^k)^\top (\mathbf{\Sigma}_{k,h}^{l_h^k})^{-1} \boldsymbol{\phi}(s_h^k, a_h^k)} \le 2^{-l_h^k}$ and $l_h^k \le l_{h-1}^k - 1$ **do**
20:             $l_h^k \leftarrow l_h^k + 1$
21:         **end while**
22:         Add element $k$ to the set $\mathcal{C}_h^{l_h^k}$
23:         Receive the reward $r_h(s_h^k, a_h^k)$ and the next state $s_{h+1}^k$
24:     **end for**
25:     Set the total level $S_{k+1}$ as $S_{k+1} = \max_{l:|\mathcal{C}_1^l|>0} l$
26: **end for**

most $K$, which incurs an additional factor of $K$ in the computational complexity in the worst case. However, if we assume the norm of $\phi(s, a)$ equals 1, then the number of levels in the first $K$ episodes is $O(\log K)$. Thus, since the computational complexity of our algorithm at each level can be bounded by that of LSVI-UCB, the computational complexity of Algorithm 2 is $O(d^2 AHK^2 \log K)$ under the above assumption.

## 6 Proof Outline

In this section, we show the proof roadmap for Theorem 5.1, which consists of three key steps.

**Step 1: Linear function approximates the optimal value function well**

We first show that with high probability, for each level $l$, our constructed linear function $\langle \mathbf{w}_{k,h}^l, \phi(s, a) \rangle$ is indeed a "good" estimation of the optimal action-value function $Q_h^*(s, a)$. By the uniform self-normalized concentration inequality over a specific function class, for any policy $\pi$, any level $l \in \mathbb{N}$ and any state-action pair $(s, a) \in \mathcal{S} \times \mathcal{A}$, we have the following concentration property:

$$(\mathbf{w}_{k,h}^l)^\top \phi(s, a) - Q_h^\pi(s, a) = \big[ \mathbb{P}_h (V_{k,h+1}^l - V_{h+1}^\pi) \big](s, a) + \Delta,$$

where $|\Delta| \le \beta_l \sqrt{\phi(s, a)^\top (\Sigma_{k,h}^l)^{-1} \phi(s, a)}$. Then, taking a backward induction for each stage $h \in [H]$, let $\Omega = \big\{ Q_{k,h}^l(s, a) \ge Q_h^*(s, a), V_{k,h}^l(s) \ge V_h^*(s), \forall (s, a) \in \mathcal{S} \times \mathcal{A}, k, l \in \mathbb{N}, h \in [H] \big\}$ denote the event that the estimated value function $Q_{k,h}^l$ and $V_{k,h}^l$ upper bounds the optimal value function $Q_h^*$ and $V_h^*$. We can show that event $\Omega$ holds with high probability. (More details can be found in Lemmas C.4 and C.5)

**Step 2: Approximation error decomposition**

On the event $\Omega$, the sub-optimality gap in round $k$ is upper bounded by the function value gap between our estimated function $Q_{k,1}^l$ and the value function of our policy $\pi_k$.

$$V_1^*(s_1^k) - V_1^{\pi_k}(s_1^k) \le \max_a \min_{1 \le l \le l_0^k - 1} Q_{k,1}^l(s_1^k, a) - Q_1^{\pi_k}(s_1^k, a_1^k) \le Q_{k,1}^{l_1^k-1}(s_1^k, a_1^k) - Q_1^{\pi_k}(s_1^k, a_1^k),$$

From now we only focus on the function value gap for level $l_h^k$. Some elementary calculation gives us

$$
\begin{aligned}
&Q_{k,h}^{l_h^k-1}(s_h^k, a_h^k) - Q_h^{\pi_k}(s_h^k, a_h^k) \\
&\le \underbrace{2\beta_{l_h^k-1}\sqrt{\phi(s_h^k, a_h^k)^\top (\Sigma_{k,h}^{l_h^k-1})^{-1}\phi(s_h^k, a_h^k)}}_{I_h^k} + Q_{k,h+1}^{l_{h+1}^k-1}(s_{h+1}^k, a_{h+1}^k) - Q_{h+1}^{\pi_k}(s_{h+1}^k, a_{h+1}^k) \\
&\quad + \underbrace{\left[\mathbb{P}_h(V_{k,h+1}^{l_h^k-1} - V_{h+1}^{\pi_k})\right](s_h^k, a_h^k) - \left(V_{k,h+1}^{l_h^k-1}(s_{h+1}^k) - V_{h+1}^{\pi_k}(s_{h+1}^k)\right)}_{\Delta_{k,h}}.
\end{aligned}
\tag{6.1}
$$

Therefore, by telescoping (6.1) from stage $h = 1$ to $H$, we conclude that the sub-optimality gap $V_1^*(s_1^k) - V_1^{\pi_k}(s_1^k)$ is upper bounded by the summation of the bonus $I_h^k$ and $\Delta_{k,h}$. The summation of the bonus $\sum I_h^k$ is the dominating error term. According to the rule of level $l$, if $k \in \mathcal{C}_h^l$ at stage $h \in [H]$, then $I_h^k$ satisfies $\sqrt{\phi(s_h^k, a_h^k)^\top (\Sigma_{k,h}^{l-1})^{-1}\phi(s_h^k, a_h^k)} \le 2^{-(l-1)}$. Furthermore, the number of elements added into set $\mathcal{C}_h^l$ can be upper bounded by $|\mathcal{C}_h^l| \le 17dlH4^l$ (See Lemma C.1). Thus we can bound the summation of $I_h^k$. For $\sum \Delta_{k,h}$, it is worth noting that $\Delta_{k,h}$ forms a martingale difference sequence, therefore by the standard Azuma-Hoeffding inequality, $\sum \Delta_{k,h}$ can be bounded by some non-dominating terms. Both of these two bounds will be used in the next step.

**Step 3: From upper confidence bonus to uniform-PAC sample complexity**

In Step 2 we have already bounded the sub-optimality gap by the summation of bonus terms. In this step, we show how to transform the gap into the final uniform-PAC sample complexity. Instead of studying any accuracy $\epsilon$ directly, we focus on a special case where $\epsilon = H/2^i (i \in \mathbb{N})$, which can be easily generalized to the general case. For each fixed $\epsilon = H/2^i (i \in \mathbb{N})$, let $\mathcal{K}$ denote the set $\mathcal{K} = \{k | V_1^*(s_1^k) - V_1^{\pi_k} \ge \epsilon\}$ and $m = |\mathcal{K}|$. On the one hand, according to the definition of set $\mathcal{K}$, the summation of regret in episode $k (k \in \mathcal{K})$ is lower bounded by $m\epsilon$. On the other hand, according to Step 2, the summation of sub-optimality gaps of episode $k (k \in \mathcal{K})$, is upper bound by

$$\sum_{k \in \mathcal{K}}[V_1^*(s_1^k) - V_1^{\pi_k}] \le \underbrace{\sum_{k \in \mathcal{K}}\sum_{h=1}^H 2\beta_{l_h^k-1}2^{-(l_h^k-1)}}_{J_1} + \underbrace{\sum_{k \in \mathcal{K}}\sum_{h=1}^H \Delta_{k,h}}_{J_2}. \tag{6.2}$$

To further bound $J_1$, we divide those episode-stage pairs $(k, h) \in \mathbb{N} \times [H]$ into two categories: $\mathcal{S}_1 = \{(k, h) | 2\beta_{l_h^k-1}2^{-(l_h^k-1)} \le \epsilon/(2H)\}$ and $\mathcal{S}_2 = \{(k, h) | 2\beta_{l_h^k-1}2^{-(l_h^k-1)} > \epsilon/(2H)\}$. For the first category $\mathcal{S}_1$, the summation of terms $I_h^k$ in this category is upper bound by

$$\sum_{k \in \mathcal{K}}\sum_{h=1}^H \mathbb{1}\{(k, h) \in \mathcal{S}_1\}2\beta_{l_h^k-1}2^{-(l_h^k-1)} \le \sum_{k \in \mathcal{K}}\sum_{h=1}^H \frac{\epsilon}{2H} = \frac{m\epsilon}{2}. \tag{6.3}$$

For any episode-stage pair $(k, h)$ in the second category $\mathcal{S}_2$, the level $l_h^k$ satisfies $2^{l_h^k} \le \widetilde{O}(dH^2/\epsilon)$ due to the choice of $\beta_{l_h^k-1}$. Suppose $l'$ is the maximum level that satisfies $2^l \le \widetilde{O}(dH^2/\epsilon)$ and for each level $l \le l'$, the cardinality of set $\mathcal{C}_h^l$ can be upper bounded by $|\mathcal{C}_h^l| \le 17dlH4^l$. Thus, the summation of terms $I_h^k$ in category $\mathcal{S}_2$ is upper bound by

$$\sum_{k \in \mathcal{K}}\sum_{h=1}^H \mathbb{1}\{(k, h) \in \mathcal{S}_2\}2\beta_{l_h^k-1}2^{-(l_h^k-1)} \le \sum_{k \in \mathcal{K}}\sum_{h=1}^H \sum_{l=1}^{l'} \mathbb{1}\{l_h^k = l\}2\beta_{l-1}2^{-(l-1)}$$

$$= \sum_{h=1}^{H} \sum_{l=1}^{l'} 2\beta_{l-1} 2^{-(l-1)} \sum_{k \in \mathcal{K}} \mathbb{1}\{l_h^k = l\}$$

$$\leq \sum_{h=1}^{H} \sum_{l=1}^{l'} 2\beta_{l-1} 2^{-(l-1)} 17 dl H 4^l$$

$$= \widetilde{O}(d^3 H^5 / \epsilon). \tag{6.4}$$

Back to (6.2), for the second term $J_2$, according to Azuma–Hoeffding inequality, it can be controlled by $\widetilde{O}(H\sqrt{Hm})$. Therefore, combining (6.3), (6.4) with the bound of $J_2$, we have

$$m\epsilon \leq \sum_{k \in \mathcal{K}} V_1^*(s_1^k) - V_1^{\pi_k} \leq m\epsilon/2 + \widetilde{O}(d^3 H^5 / \epsilon) + \widetilde{O}(H\sqrt{Hm}),$$

and it implies that the number of episodes with a sub-optimality gap greater than $\epsilon$ is bounded by $\widetilde{O}(d^3 H^5 / \epsilon^2)$. This completes the proof.

# 7    Conclusion and Future Work

In this work, we proposed two novel uniform-PAC algorithms for linear bandits and RL with linear function approximation, with the nearly state-of-the-art sample complexity. To the best of our knowledge, these are the very first results to show that linear bandits and RL with linear function approximation can also achieve uniform-PAC guarantees, similar to the tabular RL setting. We leave proving their corresponding lower bounds and proposing algorithms with near-optimal uniform-PAC sample complexity as future work.

## Acknowledgments and Disclosure of Funding

We thank the anonymous reviewers for their helpful comments. Part of this work was done when JH, DZ and QG participated the Theory of Reinforcement Learning program at the Simons Institute for the Theory of Computing in Fall 2020. JH, DZ and QG are partially supported by the National Science Foundation CAREER Award 1906169, IIS-1904183, BIGDATA IIS-1855099 and AWS Machine Learning Research Award. The views and conclusions contained in this paper are those of the authors and should not be interpreted as representing any funding agencies.

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
