# A  OFUL Algorithm is not Uniform-PAC

In this section, we consider a variant of the OFUL algorithm [1]. Then we will present a hard-to-learn linear bandit instance and show that the variant of OFUL algorithm cannot have the uniform-PAC guarantee for this instance.

In the original OFUL algorithm [1], following their notation, the agent selects the action by $\mathbf{x}_k = \operatorname{argmax}_{(\mathbf{x},\boldsymbol{\theta}) \in \mathcal{D}_k \times \Theta_{k-1}} \langle \mathbf{x}, \boldsymbol{\theta} \rangle$. Here we consider a variant of OFUL, where the agent selects the action by $\mathbf{x}_k = \operatorname{argmax}_{(\mathbf{x},\boldsymbol{\theta}) \in \mathcal{D}_k \times \Theta_{k-1} \cap B(1)} \langle \mathbf{x}, \boldsymbol{\theta} \rangle$, where $B(1)$ is a unit ball centered at zero.

We consider a special contextual linear bandit instance with dimension $d = 2$, $\boldsymbol{\theta}^* = (0, 1)$, and zero noise. The action set in the first $K$ ($K$ is an arbitrary parameter that can be chosen later) rounds is $\{(1, 0), (-1, 0)\}$ and the action set in the following $\log K$ rounds is $\{(0, 1), (0, -1)\}$. So the reward in each step can only be 1 or $-1$. The agent will randomly choose one action if both actions attain $\operatorname{argmax}_{(\mathbf{x},\boldsymbol{\theta}) \in \mathcal{D}_k \times \Theta_{k-1} \cap B(1)} \langle \mathbf{x}, \boldsymbol{\theta} \rangle$. We can show that, in the first $K$ round, the confidence radius increases since the determinant of the covariance matrix increases, and it will not provide any information about the second dimension of the vector $\boldsymbol{\theta}^*$ since the two actions are orthogonal to $\boldsymbol{\theta}^* = (0, 1)$. After the first $K$ rounds, the confidence radius will be in the order of $\log K$, and the covariance matrix $\boldsymbol{\Sigma}_K$ is a diagonal matrix and in the order of $\operatorname{diag}(K, \log K)$. We can show that both $\boldsymbol{\theta} = (0, 1)$ and $\boldsymbol{\theta} = (0, -1)$ belong to $\Theta_{k-1} \cap B(1)$, and thus attain the maximum of $\operatorname{argmax}_{(\mathbf{x},\boldsymbol{\theta}) \in \mathcal{D}_k \times \Theta_{k-1} \cap B(1)} \langle \mathbf{x}, \boldsymbol{\theta} \rangle$. Therefore, the agent will almost 'randomly' pick one of the two actions in the later $\log K$ rounds. The random selection leads to a 1-suboptimality gap for about half of the $\log K$ rounds, which indicates that OFUL cannot be uniform-PAC for any finite $f(\epsilon, \delta)$ on this bandit problem, by selecting $\log K > f(\epsilon, \delta)$.

The above reasoning can be extended to the original OFUL algorithm with a more involved argument.

# B  Proof for Theorem 4.1

In this section, we provide the proof of Theorems 4.1 and for simplicity, let $\mathcal{C}_k^l$ denote the index set $\mathcal{C}^l$ at the beginning of round $k$. We first propose the following lemmas.

**Lemma B.1.** Suppose $\lambda \geq 1$, then for each level $l \in \mathbb{N}$ and round $k \in \mathbb{N}$, the number of elements in the index set $\mathcal{C}_k^l$ is upper bounded by

$$|\mathcal{C}_k^l| \leq 17 d l 4^l.$$

*Proof.* See Appendix D.1. $\qquad\square$

Lemma B.1 suggests that $\mathcal{C}^l$ is always a finite set.

**Lemma B.2.** If we set $\lambda = 1$ and $\beta_l = 6\sqrt{dl \log(dl/\delta)}$ for every level $l \in \mathbb{N}$, then with probability at least $1 - \delta$, for all level $l \in \mathbb{N}$ and all round $k \in \mathbb{N}$, we have

$$\|\mathbf{w}_k^l - \boldsymbol{\mu}^*\|_{\boldsymbol{\Sigma}_k^l} \leq \beta_l.$$

*Proof.* See Appendix D.2. $\qquad\square$

For simplicity, let $\mathcal{E}$ denotes the event that the conclusion of Lemma B.2 holds. Therefore, Lemma B.2 suggests $\Pr(\mathcal{E}) \geq 1 - \delta$.

*Proof of Theorem 4.1.* On the event $\mathcal{E}$, for all level $l \in \mathbb{N}$, round $k \in \mathbb{N}$ and action $\mathbf{x} \in \mathcal{D}_k$, we have

$$
\begin{aligned}
(\mathbf{w}_k^l)^\top \mathbf{x} + \beta_l \sqrt{\mathbf{x}^\top (\boldsymbol{\Sigma}_k^l)^{-1} \mathbf{x}} - \langle \boldsymbol{\mu}^*, \mathbf{x} \rangle &= (\mathbf{w}_k^l - \boldsymbol{\mu}^*)^\top \mathbf{x} + \beta_l \sqrt{\mathbf{x}^\top (\boldsymbol{\Sigma}_k^l)^{-1} \mathbf{x}} \\
&\geq \beta_l \sqrt{\mathbf{x}^\top (\boldsymbol{\Sigma}_k^l)^{-1} \mathbf{x}} - \|\mathbf{w}_k^l - \boldsymbol{\mu}^*\|_{\boldsymbol{\Sigma}_k^l} \|\mathbf{x}\|_{(\boldsymbol{\Sigma}_k^l)^{-1}} \\
&\geq \beta_l \sqrt{\mathbf{x}^\top (\boldsymbol{\Sigma}_k^l)^{-1}} - \beta_l \sqrt{\mathbf{x}^\top (\boldsymbol{\Sigma}_k^l)^{-1}} \\
&= 0,
\end{aligned}
\tag{B.1}
$$

where the first inequality holds due to Cauchy-Schwarz inequality and the second inequality holds due to the definition of event $\mathcal{E}$. (B.1) implies that the estimated reward for each action $\mathbf{x} \in \mathcal{D}_k$ at level $l$: $(\mathbf{w}_k^l)^\top \mathbf{x} + \beta_l \sqrt{\mathbf{x}^\top (\boldsymbol{\Sigma}_k^l)^{-1} \mathbf{x}}$ is an upper confidence bound of the expected reward $\langle \boldsymbol{\mu}^*, \mathbf{x} \rangle$. Thus, for each action $\mathbf{x} \in \mathcal{D}_k$, we have

$$\min_{1 \leq l \leq S_k} (\mathbf{w}_k^l)^\top \mathbf{x} + \beta_l \sqrt{\mathbf{x}^\top (\boldsymbol{\Sigma}_k^l)^{-1} \mathbf{x}} \geq \min_{1 \leq l \leq S_k} \langle \boldsymbol{\mu}^*, \mathbf{x} \rangle = \langle \boldsymbol{\mu}^*, \mathbf{x} \rangle. \tag{B.2}$$

Therefore, for the sub-optimality gap at round $k$, we have

$$\max_{\mathbf{x} \in \mathcal{D}_k} \langle \boldsymbol{\mu}^*, \mathbf{x} \rangle - \langle \boldsymbol{\mu}^*, \mathbf{x}_k \rangle \leq \max_{\mathbf{x} \in \mathcal{D}_k} \min_{1 \leq l \leq S_k} (\mathbf{w}_k^l)^\top \mathbf{x} + \beta_l \sqrt{\mathbf{x}^\top (\boldsymbol{\Sigma}_k^l)^{-1} \mathbf{x}} - \langle \boldsymbol{\mu}^*, \mathbf{x}_k \rangle$$

$$= \min_{1 \leq l \leq S_k} (\mathbf{w}_k^l)^\top \mathbf{x}_k + \beta_l \sqrt{\mathbf{x}_k^\top (\boldsymbol{\Sigma}_k^l)^{-1} \mathbf{x}_k} - \langle \boldsymbol{\mu}^*, \mathbf{x}_k \rangle, \tag{B.3}$$

where the first inequality holds due to (B.2) and the second equality holds due to the policy in Algorithm 1 (line 8). Thus, for each round $k \in \mathbb{N}$, if the level $l_k > 1$, we have

$$\max_{\mathbf{x} \in \mathcal{D}_k} \langle \boldsymbol{\mu}^*, \mathbf{x} \rangle - \langle \boldsymbol{\mu}^*, \mathbf{x}_k \rangle \leq (\mathbf{w}_k^{l_k-1})^\top \mathbf{x}_k + \beta_{l_k-1} \sqrt{\mathbf{x}_k^\top (\boldsymbol{\Sigma}_k^{l_k-1})^{-1} \mathbf{x}_k} - \langle \boldsymbol{\mu}^*, \mathbf{x}_k \rangle$$

$$= (\mathbf{w}_k^{l_k-1} - \boldsymbol{\mu}^*)^\top \mathbf{x}_k + \beta_{l_k-1} \sqrt{\mathbf{x}_k^\top (\boldsymbol{\Sigma}_k^{l_k-1})^{-1} \mathbf{x}_k}$$

$$\leq \|\mathbf{w}_k^{l_k-1} - \boldsymbol{\mu}^*\|_{\boldsymbol{\Sigma}_k^{l_k-1}} \|\mathbf{x}_k\|_{(\boldsymbol{\Sigma}_k^{l_k-1})^{-1}} + \beta_{l_k-1} \sqrt{\mathbf{x}_k^\top (\boldsymbol{\Sigma}_k^{l_k-1})^{-1} \mathbf{x}_k}$$

$$\leq 2\beta_{l_k-1} \sqrt{\mathbf{x}_k^\top (\boldsymbol{\Sigma}_k^{l_k-1})^{-1} \mathbf{x}_k}$$

$$\leq 2\beta_{l_k-1} \times 2^{-(l_k-1)},$$

where the first inequality holds due to (B.3) with the fact that $l_k - 1 \leq S_k$, the second inequality holds due to Cauchy-Schwarz inequality, the third inequality holds due to the definition of event $\mathcal{E}$ and the last inequality holds due to the definition of level $l_k$ in Algorithm 1 (line 10 to line 11). Since we set the parameter $\beta_l = 6\sqrt{dl \log(dl/\delta)}$, there exists a large constant $C$ such that for any level $l$ satisfied $2^l \geq C\sqrt{d \log^2(d/(\delta\epsilon))}/\epsilon$, we have $2\beta_{l-1} \times 2^{-(l-1)} \leq \epsilon$. For simplicity, we denote the minimum level $m = \left\lceil \log\left( C\sqrt{d \log^2(d/(\delta\epsilon))}/\epsilon \right) \right\rceil$. Then for each round $k$, if level $l_k > m$, we have

$$\max_{\mathbf{x} \in \mathcal{D}_k} \langle \boldsymbol{\mu}^*, \mathbf{x} \rangle - \langle \boldsymbol{\mu}^*, \mathbf{x}_k \rangle \leq 2\beta_{l_k-1} \times 2^{-(l_k-1)} \leq \epsilon. \tag{B.4}$$

Thus, for any $\epsilon > 0$, we have

$$\sum_{k=1}^\infty \mathbb{1}\left\{ \max_{\mathbf{x} \in \mathcal{D}_k} \langle \boldsymbol{\mu}^*, \mathbf{x} \rangle - \langle \boldsymbol{\mu}^*, \mathbf{x}_k \rangle > \epsilon \right\} \leq \sum_{k=1}^\infty \mathbb{1}\{l_k \leq m\}$$

$$= \sum_{k=1}^\infty \sum_{l=1}^m \mathbb{1}\{l_k = l\}$$

$$= \sum_{l=1}^m \sum_{k=1}^\infty \mathbb{1}\{l_k = l\},$$

where the inequality holds due to (B.4). According to Lemma B.1, the number of rounds with sub-optimality more than $\epsilon$ can be further bounded by

$$\sum_{k=1}^\infty \mathbb{1}\left\{ \max_{\mathbf{x} \in \mathcal{D}_k} \langle \boldsymbol{\mu}^*, \mathbf{x} \rangle - \langle \boldsymbol{\mu}^*, \mathbf{x}_k \rangle > \epsilon \right\} \leq \sum_{l=1}^m \sum_{k=1}^\infty \mathbb{1}\{l_k = l\}$$

$$\leq \sum_{l=1}^m 17dl4^l$$

$$\leq C'd^2 \log^3(d/(\delta\epsilon))/\epsilon^2,$$

where the second inequality holds due to Lemma B.1 and the last inequality holds due to the definition of $m$ with the fact that $\sum_{l=1}^m l4^l \leq m4^{m+1}$. Thus, we finish the proof of Theorem 4.1. $\qquad\square$

# C    Proof of Theorem 5.1

In this section, we provide the proof of Theorems 5.1 and for simplicity, let $\mathcal{C}_{k,h}^l$ denote the index set $\mathcal{C}_h^l$ at the beginning of episode $k$. We first propose the following lemmas.

**Lemma C.1.** Suppose the parameter $\lambda$ satisfies $\lambda \geq 1$, then for each level $l \in \mathbb{N}$ each stage $h \in [H]$ and each episode $k \in \mathbb{N}$, the number of elements in the set $\mathcal{C}_{k,h}^l$ is upper bounded by

$$|\mathcal{C}_{k,h}^l| \leq 17dlh4^l.$$

*Proof.* See Appendix E.1. □

**Lemma C.2.** Under Assumption 3.1, for each stage $h \in [H]$, each level $l \in \mathbb{N}$ and each episode $k \in \mathbb{N}$, the norm of weight vector $\mathbf{w}_{k,h}^l$ can be upper bounded by

$$\|\mathbf{w}_{k,h}^l\|_2 \leq \frac{9d2^l\sqrt{H^3l}}{\sqrt{\lambda}}.$$

*Proof.* See Appendix E.2. □

**Lemma C.3.** Suppose the parameter $\lambda = 1$, then there exists a large constant $C$, such that with probability $1 - \delta/2$, for all stage $h \in [H]$, all episode $k \in \mathbb{N}$ and all level $l \in \mathbb{N}$, we have

$$\left\| \sum_{i \in \mathcal{C}_{k,h}^l} \phi(s_h^i, a_h^i)\left[ V_{k,h+1}^l(s_{h+1}^i) - [\mathbb{P}_h V_{k,h+1}^l](s_h^i, a_h^i) \right] \right\|_{(\mathbf{\Sigma}_{k,h}^l)^{-1}} \leq CdHl\sqrt{\log(dlH\beta_l^2/\delta)}.$$

*Proof.* See Appendix E.3. □

For simplicity, let $\mathcal{E}$ denote the event that the conclusion of Lemma C.3 holds. Therefore, Lemma C.3 shows that $\Pr(\mathcal{E}) \geq 1 - \delta/2$.

**Lemma C.4.** Suppose $\lambda = 1$ and $\beta_l = CdHl\sqrt{\log(dlH/\delta)}$ with a large constant $C$, then on the event $\mathcal{E}$, for all state-action pair $(s,a) \in \mathcal{S} \times \mathcal{A}$, stage $h \in [H]$, episode $k \in \mathbb{N}$, level $l \in \mathbb{N}$ and any policy $\pi$, we have

$$(\mathbf{w}_{k,h}^l)^\top \phi(s,a) - Q_h^\pi(s,a) = \left[ \mathbb{P}_h(V_{k,h+1}^l - V_{h+1}^\pi) \right](s,a) + \Delta,$$

where $|\Delta| \leq \beta_l\sqrt{\phi(s,a)^\top(\Sigma_{k,h}^l)^{-1}\phi(s,a)}$.

*Proof.* See Appendix E.4. □

**Lemma C.5.** On the event $\mathcal{E}$, for all state-action pair $(s,a) \in \mathcal{S} \times \mathcal{A}$, stage $h \in [H]$, episode $k \in \mathbb{N}$ and level $l \in \mathbb{N}$, we have

$$Q_{k,h}^l(s,a) \geq Q_h^*(s,a), V_{k,h}^l(s) \geq V_h^*(s).$$

*Proof.* See Appendix E.5. □

Now we begin to prove Theorem 5.1.

*Proof of Theorem 5.1.* Firstly, we focus on the special case that $\epsilon = H2^{-i}(i \in \mathbb{N})$. Since $\epsilon = H2^{-i}$ and we set the parameter $\beta_l = CdHl\sqrt{\log(dlH/\delta)}$, there exists a large constant $C'$ such that for any level $l$ satisfied $2^l \geq C'dH^2\log^{1.5}(dH/(\delta\epsilon))/\epsilon$, we have

$$4\beta_{l-1}2^{-l} = CdHl\sqrt{\log(dlH\beta_{l-1}^2/\delta)}2^{-l} \leq \epsilon/(2H).$$

For simplicity, we denote the maximum level $l'$ as $l' = \left\lceil \log\left( C'dH^2\log^{1.5}(dH/(\delta\epsilon))/\epsilon \right) \right\rceil$.

Now, let $k_0 = 0$, and for each $i \in \mathbb{N}$, we denote $k_i$ as the minimum index of the episode where the sub-optimality gap is more than $\epsilon$, such that

$$k_i = \min\left\{k : k > k_{i-1}, V_1^*(s_1^k) - V_1^{\pi_k}(s_1^k) \geq \epsilon\right\}. \tag{C.1}$$

Now, we denote the set $K = \{k_i : i \in \mathbb{N}, k_i < +\infty\}$ and we assume $K = \{k_1, .., k_m\}$. According to the definition of $k_i$ in (C.1), we have

$$\sum_{i=1}^m V_1^*(s_1^{k_i}) - V_1^{\pi_{k_i}}(s_1^{k_i}) \geq m\epsilon. \tag{C.2}$$

On the other hand, for each episode $k \in \mathbb{N}$ with total level $S_k$ at the beginning of episode $k$, we have

$$
\begin{aligned}
V_1^*(s_1^k) - V_1^{\pi_k}(s_1^k) &= \max_a Q_1^*(s_1^k, a) - Q_1^{\pi_k}(s_1^k, a_1^k) \\
&\leq \max_a \min_{1 \leq l \leq S_k} Q_{k,1}^l(s_1^k, a) - Q_1^{\pi_k}(s_1^k, a_1^k) \\
&= \min_{1 \leq l \leq S_k} Q_{k,1}^l(s_1^k, a_1^k) - Q_1^{\pi_k}(s_1^k, a_1^k) \\
&\leq Q_{k,1}^{l_1^k - 1}(s_1^k, a_1^k) - Q_1^{\pi_k}(s_1^k, a_1^k), 
\end{aligned} \tag{C.3}
$$

where the first inequality holds due to Lemma C.5, the third equation holds due to the policy in Algorithm 2 and the last inequality holds due to the fact that $l_1^k - 1 \leq S_k$. Furthermore, for each stage $h \in [H]$ and each episode $k \in \mathbb{N}$, we have

$$
\begin{aligned}
&Q_{k,h}^{l_h^k-1}(s_h^k, a_h^k) - Q_h^{\pi_k}(s_h^k, a_h^k) \\
&\leq (\mathbf{w}_{k,h}^{l_h^k-1})^\top \phi(s_h^k, a_h^k) + \beta_{l_h^k-1}\sqrt{\phi(s_h^k, a_h^k)^\top (\mathbf{\Sigma}_{k,h}^{l_h^k-1})^{-1}\phi(s, a)} - Q_h^{\pi_k}(s_h^k, a_h^k) \\
&\leq 2\beta_{l_h^k-1}\sqrt{\phi(s_h^k, a_h^k)^\top (\mathbf{\Sigma}_{k,h}^{l_h^k-1})^{-1}\phi(s_h^k, a_h^k)} + \left[\mathbb{P}_h(V_{k,h+1}^l - V_{h+1}^{\pi_k})\right](s_h^k, a_h^k) \\
&\leq 2\beta_{l_h^k-1}2^{-(l_h^k-1)} + \left[\mathbb{P}_h(V_{k,h+1}^{l_h^k-1} - V_{h+1}^{\pi_k})\right](s_h^k, a_h^k) \\
&= 4\beta_{l_h^k-1}2^{-l_h^k} + \underbrace{\left[\mathbb{P}_h(V_{k,h+1}^{l_h^k-1} - V_{h+1}^{\pi_k})\right](s_h^k, a_h^k) - \left(V_{k,h+1}^{l_h^k-1}(s_{h+1}^k) - V_{h+1}^{\pi_k}(s_{h+1}^k)\right)}_{\Delta_{k,h}} \\
&\quad + V_{k,h+1}^{l_h^k-1}(s_{h+1}^k) - V_{h+1}^{\pi_k}(s_{h+1}^k),
\end{aligned} \tag{C.4}
$$

where the first inequality holds due to the definition of value function $Q_{k,h}^l$ in Algorithm 2, the second inequality holds due to Lemma C.4 and the last inequality holds due to the definition of level $l_h^k$ in Algorithm 2. Furthermore, for the term $V_{k,h+1}^{l_h^k-1}(s_{h+1}^k)$, it can be upper bounded by

$$
\begin{aligned}
V_{k,h+1}^{l_h^k-1}(s_{h+1}^k) &= \max_a \min_{1 \leq l \leq l_h^k-1} Q_{k,h+1}^l(s_{h+1}^k, a) \\
&= \min_{1 \leq l \leq l_h^k-1} Q_{k,h+1}^l(s_{h+1}^k, a_{h+1}^k) \\
&\leq Q_{k,h+1}^{l_{h+1}^k-1}(s_{h+1}^k, a_{h+1}^k), 
\end{aligned} \tag{C.5}
$$

where the inequality holds due to the fact that $l_{h+1}^k - 1 \leq l_h^k - 1$. Substituting (C.5) in to (C.4) and taking a summation of (C.4) with all stage $h \in [H]$, we have

$$V_1^*(s_1^k) - V_1^{\pi_k}(s_1^k) \leq Q_{k,1}^{l_1^k-1}(s_1^k, a_1^k) - Q_1^{\pi_k}(s_1^k, a_1^k) \leq \sum_{h=1}^H 4\beta_{l_h^k-1}2^{-l_h^k} + \sum_{h=1}^H \Delta_{k,h}. \tag{C.6}$$

Taking a summation of (C.6) over all episode $k_i \in K$, we have

$$\sum_{i=1}^m V_1^*(s_1^{k_i}) - V_1^{\pi_{k_i}}(s_1^{k_i}) \leq \underbrace{\sum_{i=1}^m \sum_{h=1}^H 4\beta_{l_h^{k_i}-1}2^{-l_h^{k_i}}}_{I_1} + \underbrace{\sum_{i=1}^m \sum_{h=1}^H \Delta_{k_i,h}}_{I_2}. \tag{C.7}$$

Since $4\beta_{l-1}2^{-l} \le \epsilon/(2H)$ holds for all level $l > l'$, the term $I_1$ can be upper bounded by

$$
\begin{aligned}
I_1 &= \sum_{i=1}^{m}\sum_{h=1}^{H} 4\beta_{l_h^{k_i}-1} 2^{-l_h^{k_i}} \\
&\le \sum_{i=1}^{m}\sum_{h=1}^{H} \left( \mathbb{1}\{l_h^{k_i} \le l'\} 4\beta_{l_h^{k_i}-1} 2^{-l_h^{k_i}} + \frac{\epsilon}{2H} \right) \\
&= \sum_{i=1}^{m}\sum_{h=1}^{H} \left( \sum_{l=1}^{l'} \mathbb{1}\{l_h^{k_i}=l\} 4\beta_{l-1}2^{-l} + \frac{\epsilon}{2H} \right) \\
&= \frac{m\epsilon}{2} + \sum_{h=1}^{H}\sum_{l=1}^{l'} 4\beta_{l-1}2^{-l}\sum_{i=1}^{m}\mathbb{1}\{l_h^{k_i}=l\}.
\end{aligned}
$$

According to Lemma C.1, the number of elements added into the set $\mathcal{C}_h^l$ is upper bounded by $|\mathcal{C}_h^l| \le 17dlh4^l$ and it implies that $\sum_{i=1}^m \mathbb{1}\{l_h^{k_i}=l\} \le \sum_{k=1}^{+\infty}\mathbb{1}\{l_h^k = l\} \le 17dlh4^l$. Thus, we have

$$
\begin{aligned}
I_1 &\le \frac{m\epsilon}{2} + \sum_{h=1}^{H}\sum_{l=1}^{l'} 4\beta_{l-1}2^{-l}\sum_{i=1}^{m}\mathbb{1}\{l_h^{k_i}=l\} \\
&\le \frac{m\epsilon}{2} + \sum_{h=1}^{H}\sum_{l=1}^{l'} 4\beta_{l-1}2^{-l}\times 17dlh4^l \\
&\le \frac{m\epsilon}{2} + 136\beta_{l'-1}2^{l'}dl'H^2. \tag{C.8}
\end{aligned}
$$

According to the definition of level $l'$ and parameter $\beta_{l'}$, there exist a large constant $C''$ such that $I_1 \le m\epsilon/2 + C''d^3H^5\log^4(dH/(\delta\epsilon))/\epsilon$.

For the term $I_2$, according to Lemma F.1, for any fixed number $n \in \mathbb{N}$ and $\epsilon = H/2^i$, with probability at least $1 - \delta/\big(2i(i+1)n(n+1)\big)$, we have

$$
\sum_{i=1}^{n}\sum_{h=1}^{H} \Delta_{k_i,h} \le H\sqrt{2Hn\log\frac{2i(i+1)n(n+1)}{\delta}}.
$$

Therefore, taking a union bound, with probability at least $1 - \delta/\big(2i(i+1)\big)$, for all $n \in \mathbb{N}$, we have

$$
\sum_{i=1}^{n}\sum_{h=1}^{H} \Delta_{k_i,h} \le H\sqrt{2Hn\log\frac{2i(i+1)n(n+1)}{\delta}}.
$$

Thus, for the term $I_2$ and $\epsilon = H/2^i$, with probability at least $1 - \delta/\big(2i(i+1)\big)$, we have

$$
I_2 = \sum_{i=1}^{m}\sum_{h=1}^{H} \Delta_{k_i,h} \le H\sqrt{2Hm\log\frac{2i(i+1)m(m+1)}{\delta}}. \tag{C.9}
$$

Substituting (C.8) and (C.9) into (C.7), for $\epsilon = H/2^i$, we have

$$
\begin{aligned}
m\epsilon &\ge \sum_{i=1}^{m} V_1^*(s_1^{k_i}) - V_1^{\pi_{k_i}}(s_1^{k_i}) \\
&\ge \frac{m\epsilon}{2} + C''d^3H^5\log^4(dH/(\delta\epsilon))/\epsilon \\
&\quad + H\sqrt{2m\log\frac{2i(i+1)m(m+1)}{\delta}},
\end{aligned}
$$

which implies $m \le O\big(d^3H^5\log^4\big(dH/(\delta\epsilon)\big)/\epsilon^2\big)$. Finally, taking an union bound with the event $\mathcal{E}$ and (C.9), with probability at least $1 - \delta/2 - \sum_{i=1}^{\infty}\delta/\big(2i(i+1)\big) = 1-\delta$, for all $\epsilon = H/2^i (i \in \mathbb{N})$, we have

$$
\sum_{k=1}^{\infty}\mathbb{1}\{V_1^*(s_1^k) - V_1^{\pi_k}(s_1^k) \ge \epsilon\} \le O\big(d^3H^5\log^4\big(dH/(\delta\epsilon)\big)/\epsilon^2\big).
$$

Finally, we extend the result to general $\epsilon > 0$. For any $H/2^i \leq \epsilon \leq H/2^{i-1}$, we have

$$\sum_{k=1}^{\infty} \mathbb{1}\{V_1^*(s_1^k) - V_1^{\pi_k}(s_1^k) \geq \epsilon\} \leq \sum_{k=1}^{\infty} \mathbb{1}\{V_1^*(s_1^k) - V_1^{\pi_k}(s_1^k) \geq H/2^i\}$$

$$\leq O\Big(d^3 H^5 \log^4\big(dH/(\delta\epsilon')\big)/(H/2^i)^2\Big)$$

$$= O\Big(d^3 H^5 \log^4\big(dH/(\delta\epsilon)\big)/\epsilon^2\Big).$$

Thus, we finish the proof of Theorem 5.1. $\qquad\square$

## D  Proof of Lemma in Section B

### D.1  Proof of Lemma B.1

**Lemma D.1** (Lemma 11, [1])**.** For any vector sequence $\{\mathbf{x}_k\}_{k=1}^K$ in $\mathbb{R}^d$, We denote $\boldsymbol{\Sigma}_0 = \lambda\mathbf{I}$ and $\boldsymbol{\Sigma}_k = \boldsymbol{\Sigma}_0 + \sum_{i=1}^k \mathbf{x}_i\mathbf{x}_i^\top$. If $\lambda \geq \max(1, L^2)$ and $\|\mathbf{x}_k\|_2 \leq L$ holds for all $k \in [K]$, then we have

$$\sum_{k=1}^K \|\mathbf{x}_k\|_{\boldsymbol{\Sigma}_{k-1}^{-1}}^2 \leq 2d \log \frac{d\lambda + KL^2}{d\lambda}.$$

*Proof of Lemma B.1.* We focus on round $k$ and we suppose set $\mathcal{C}_k^l = \{k_1, .., k_m\}$ at that time, where $1 \leq k_1 < k_2 < .. < k_m < k$. According to the update rule of set $\mathcal{C}^l$ in Algorithm 1 (line 9 to line 13), for each $2 \leq i \leq m$, we have $S_{k_i} \geq l$ and it implies that

$$\mathbf{x}_{k_i}^\top (\boldsymbol{\Sigma}_{k_i}^l)^{-1} \mathbf{x}_{k_i} \geq 4^{-l}, \tag{D.1}$$

where $\boldsymbol{\Sigma}_{k_i}^l = \lambda\mathbf{I} + \sum_{j=1}^{i-1} \mathbf{x}_{k_j}\mathbf{x}_{k_j}^\top$. Therefore, taking a summation for (D.1) over all $2 \leq i \leq m$, we have

$$\sum_{i=1}^m \mathbf{x}_{k_i}^\top (\boldsymbol{\Sigma}_{k_i}^l)^{-1} \mathbf{x}_{k_i} \geq \sum_{i=2}^m \mathbf{x}_{k_i}^\top (\boldsymbol{\Sigma}_{k_i}^l)^{-1} \mathbf{x}_{k_i} \geq (m-1)4^{-l}, \tag{D.2}$$

where the first inequality holds due to $\mathbf{x}_{k_1}^\top (\boldsymbol{\Sigma}_{k_1}^l)^{-1} \mathbf{x}_{k_1} \geq 0$ and the second inequality holds due to (D.1). On the other hand, according to Lemma D.1, this summation is upper bounded by

$$\sum_{i=1}^m \mathbf{x}_{k_i}^\top (\boldsymbol{\Sigma}_{k_i}^l)^{-1} \mathbf{x}_{k_i} \leq 2d \log \frac{d\lambda + m}{d\lambda} \leq 2d \log(1 + m/d), \tag{D.3}$$

where the first inequality holds due to Lemma D.1 with $\|\mathbf{x}_{k_i}\|_2 \leq 1$ and the second inequality holds due to $\lambda \geq 1$. Combining (D.2) and (D.3), we have

$$(m-1)4^{-l} \leq 2d \log(1 + m/d),$$

which implies that the size of set $|\mathcal{C}_k^l|$ is upper bounded by $|\mathcal{C}_k^l| = m \leq 17dl4^l$ for each $k \in \mathbb{N}$. Therefore, we finish the proof of Lemma B.1. $\qquad\square$

### D.2  Proof of Lemma B.2

**Lemma D.2** (Theorem 2, [1])**.** Let $\{\epsilon_t\}_{t=1}^{\infty}$ be a real-valued stochastic process with corresponding filtration $\{\mathcal{F}_t\}_{t=0}^{\infty}$ such that $\epsilon_t$ is $\mathcal{F}_t$-measure and $\epsilon_t$ is conditionally $R$-sub-Gaussian, *i.e.*

$$\forall \lambda \in \mathbb{R}, \mathbb{E}[e^{\lambda\epsilon_t}|\mathcal{F}_{t-1}] \leq \exp\left(\frac{\lambda^2 R^2}{2}\right).$$

Let $\{\mathbf{x}_t\}_{t=1}^{\infty}$ be an $\mathbb{R}^d$-valued stochastic process where $\mathbf{x}_t$ is $\mathcal{F}_{t-1}$-measurable and we define $y_t = \langle \mathbf{x}_t, \boldsymbol{\theta}^* \rangle + \epsilon_t$. With this notation, for any $t \geq 0$, we further define

$$\boldsymbol{\Sigma}_t = \lambda I + \sum_{i=1}^t \mathbf{x}_t\mathbf{x}_t^\top, \mathbf{b}_t = \sum_{i=1}^t \mathbf{x}_t y_t, \mathbf{w}_t = (\boldsymbol{\Sigma}_t)^{-1}\mathbf{b}_t.$$

If we assume $\|\boldsymbol{\theta}^*\| \le S$ and $\|\mathbf{x}_t\| \le L$ holds for all $t \in \mathbb{N}$, then with probability at least $1 - \delta$, for all $t \ge 0$, we have

$$\|\boldsymbol{\theta}^* - \mathbf{w}_t\|_{\boldsymbol{\Sigma}_t} \le R\sqrt{d \log\left(\frac{1 + tL^2/\lambda}{\delta}\right)} + \sqrt{\lambda}S.$$

*Proof of Lemma B.2.* In this proof, we first focus on a fixed level $l \in \mathbb{N}$ and then turn back to all level $l$. For a fixed level $l \in [N]$, we denote $k_0 = 0$, and for $i \in \mathbb{N}$, we denote $k_i$ as the minimum index of the round where the action is added to the set $\mathcal{C}^l$:

$$k_i = \min\{k : k > k_{i-1}, l_k = l\}. \tag{D.4}$$

Under this notation, for all round $k(k_i < k \le k_{i+1})$, we have

$$\boldsymbol{\Sigma}_k^l = \lambda \mathbf{I} + \sum_{j=1}^{i} \mathbf{x}_{k_j} \mathbf{x}_{k_j}^\top, \quad \mathbf{b}_k^l = \sum_{j=1}^{i} \mathbf{x}_{k_j} r_{k_j}, \quad \mathbf{w}_k^l = (\boldsymbol{\Sigma}_k^l)^{-1} \mathbf{b}_k^l. \tag{D.5}$$

Now, we consider the $\sigma$-algebra filtration $\mathcal{F}_i = \sigma(\mathbf{x}_1, .., \mathbf{x}_{k_{i+1}}, r_1, .., r_{k_{i+1}-1})$ that contains all randomness before receiving the reward $r_{k_{i+1}}$ at round $k_{i+1}$. By the definition of $\mathcal{F}_{i-1}$, vector $\mathbf{x}_{k_i}$ is $\mathcal{F}_{i-1}$-measurable and the noise $\epsilon_{k_i} = r_{k_i} - \langle \mathbf{x}_{k_i}, \boldsymbol{\mu}^* \rangle$ is $\mathcal{F}_i$-measurable. Since we choose the level $l_k$ and add element $k$ to the corresponding set $\mathcal{C}^{l_k}$ before receiving the reward $r_k$ at round $k$, the noise $\epsilon_{k_i}$ is conditionally 1-Sub-Gaussian. According to Lemma D.2, with probability at least $1 - \delta(l(l+1))$, for all $i \ge 0$, we have

$$\|\boldsymbol{\mu}^* - \mathbf{w}_{k_{i+1}}^l\|_{\boldsymbol{\Sigma}_{k_{i+1}}^l} \le \sqrt{d \log\left(\frac{i+1}{\delta/(l(l+1))}\right)} + 1. \tag{D.6}$$

Combining (D.5) and (D.6), for all round $k(k_i < k \le k_{i+1})$, we have

$$\|\boldsymbol{\mu}^* - \mathbf{w}_k^l\|_{\boldsymbol{\Sigma}_k^l} \le \sqrt{d \log\left(\frac{i+1}{\delta/(l(l+1))}\right)} + 1. \tag{D.7}$$

Furthermore, Lemma B.1 suggests that the size of set $|\mathcal{C}^l|$ is upper bounded by $|\mathcal{C}^l| \le 17dl4^l$, which implies that $k_{17dl4^l+1} = +\infty$. Thus, (D.7) implies that with probability at least $1 - \delta/(l(l+1))$, for all round $k \in \mathbb{N}$, we have

$$\|\boldsymbol{\mu}^* - \mathbf{w}_k^l\|_{\boldsymbol{\Sigma}_k^l} \le \sqrt{d \log\left(\frac{17dl4^l + 1}{\delta/(l(l+1))}\right)} + 1 \le \beta_l. \tag{D.8}$$

Finally, taking a union bound for (D.8) over all level $l \in \mathbb{N}$, with probability at least $1 - \sum_{l=1}^{\infty}\left(\delta/(l(l+1))\right) = 1 - \delta$, for all level $l \in [N]$ and all round $k \in \mathbb{N}$, we have

$$\|\mathbf{w}_k^l - \boldsymbol{\mu}^*\|_{\boldsymbol{\Sigma}_k^l} \le \beta_l.$$

Thus, we finish the proof of Lemma B.2 □

# E Proof of Lemma in Section C

## E.1 Proof of Lemma C.1

*Proof of Lemma C.1.* Similar to the proof of Lemma B.1, we focus on episode $k$ and we suppose set $\mathcal{C}_{k,h}^l = \{k_1, .., k_m\}$ at that time, where $1 \le k_1 < k_2 < .. < k_m < k$. For simplicity, we further define the auxiliary sets $\mathcal{B}_{k,h}^l$ as

$$\mathcal{B}_{k,h}^l = \{i | 1 \le i < k; l_h^i = l; (h = 1 \text{ or } l_h^i < l_{h-1}^i)\}.$$

Notice that for each stage $h \ge 2$ and episode $i \in [k]$, there are two stopping rules for the while loop in Algorithm 2 (line 19 to line 20) and $\mathcal{B}_{k,h}^l$ consists of all episode $i \in [k]$ that stop with the first rule.

Furthermore, for all element $k_i \in \mathcal{C}_{k,h}^l$ with the second rule stopping rule, we have $l_{h-1}^k = l_h^k = l$ and it implies that $k_i \in \mathcal{C}_{k,h-1}^l$. Combining these two cases, we have $\mathcal{C}_h^l \subseteq \mathcal{B}_{k,h}^l \cup \mathcal{C}_{k,h-1}^l$ and it implies that

$$m = |\mathcal{C}_{k,h}^l| \leq \sum_{j=1}^h |\mathcal{B}_{k,j}^l|, \tag{E.1}$$

where the inequality holds due to $|\mathcal{C} \cup \mathcal{B}| \leq |\mathcal{C}| + |\mathcal{B}|$ and the fact that $\mathcal{C}_{k,1}^l = \mathcal{B}_{k,1}^l$.

Now, we only need to control the size of $\mathcal{B}_{k,h}^l$ for each episode $k \in \mathbb{N}$. For simplicity, we suppose set $\mathcal{B}_{k,h}^l = \{k_1, .., k_n\}$, where $1 \leq k_1 \leq k_2 \leq ... \leq k_n < k$. According to the definition of level $l_h^k$ in Algorithm 2 (line 19 to line 20), for $2 \leq i \leq n$, we have

$$\phi(s_h^{k_i}, a_h^{k_i})^\top (\boldsymbol{\Sigma}_{k_i,h}^l)^{-1} \phi(s_h^{k_i}, a_h^{k_i}) \geq 4^{-l}.$$

Since $\mathcal{B}_{k,h}^l \subseteq \mathcal{C}_{k,h}^l$ holds for all stage $h \in [H]$, all level $l \in \mathbb{N}$ and all episode $k \in \mathbb{N}$, we have $\boldsymbol{\Sigma}_{k_i,h}^l \succeq \lambda \mathbf{I} + \sum_{j=1}^{i-1} \phi(s_h^{k_i}, a_h^{k_i}) \phi(s_h^{k_i}, a_h^{k_i})^\top$ and it implies that

$$\phi(s_h^{k_i}, a_h^{k_i})^\top (\boldsymbol{\Gamma}_{k_i,h}^l)^{-1} \phi(s_h^{k_i}, a_h^{k_i}) \geq \phi(s_h^{k_i}, a_h^{k_i})^\top (\boldsymbol{\Sigma}_{k_i,h}^l)^{-1} \phi(s_h^{k_i}, a_h^{k_i}) \geq 4^{-l}. \tag{E.2}$$

where $\boldsymbol{\Gamma}_{k_i,h}^l = \lambda \mathbf{I} + \sum_{j=1}^{i-1} \phi(s_h^{k_i}, a_h^{k_i}) \phi(s_h^{k_i}, a_h^{k_i})^\top$ . Thus, taking a summation for (E.2) over all $2 \leq i \leq n$, we have

$$\sum_{i=1}^n \phi(s_h^{k_i}, a_h^{k_i})^\top (\boldsymbol{\Gamma}_{k_i,h}^l)^{-1} \phi(s_h^{k_i}, a_h^{k_i}) \geq \sum_{i=2}^n \phi(s_h^{k_i}, a_h^{k_i})^\top (\boldsymbol{\Gamma}_{k_i,h}^l)^{-1} \phi(s_h^{k_i}, a_h^{k_i}) \geq (n-1)4^{-l},$$

$$\tag{E.3}$$

where the first inequality holds due to $\phi(s_h^{k_1}, a_h^{k_1})^\top (\boldsymbol{\Gamma}_{k_1,h}^l)^{-1} \phi(s_h^{k_1}, a_h^{k_1}) \geq 0$ and the second inequality holds due to (E.2). On the other hand, according to Lemma D.1, this summation is upper bounded by

$$\sum_{i=1}^n \phi(s_h^{k_i}, a_h^{k_i})^\top (\boldsymbol{\Gamma}_{k_i,h}^l)^{-1} \phi(s_h^{k_i}, a_h^{k_i}) \leq 2d \log \frac{d\lambda + n}{d\lambda} \leq 2d \log(1 + n/d), \tag{E.4}$$

where the first inequality holds due to Lemma D.1 with the fact that $\|\phi(s,a)\|_2 \leq 1$ and the second inequality holds due to $\lambda \geq 1$. Combining (E.3) and (E.4), we have

$$(n-1)4^{-l} \leq 2d \log(1 + n/d), \tag{E.5}$$

which implies that $|\mathcal{B}_{k,h}^l| = n \leq 17dl4^l$. Finally, substituting (E.5) into (E.1), we have.

$$m = |\mathcal{C}_{k,h}^l| \leq \sum_{j=1}^h |\mathcal{B}_{k,j}^l| \leq 17dhl4^l. \tag{E.6}$$

Therefore, we finish the proof of Lemma C.1. $\qquad\square$

## E.2 Proof of Lemma C.2

*Proof of Lemma C.2.* In this proof, we only need to show that the norm of vector $\mathbf{w}_{k,h}^l$ is bounded for each fixed episode $k \in \mathbb{N}$ and fixed level $l \in \mathbb{N}$. For simplicity, let $\mathcal{C}_{k,h}^l = \{k_1, .., k_m\}$ denote the index set $\mathcal{C}_h^l$ at the beginning of episode $k$, where $1 \leq k_1 < k_2 < .. < k_m < k$. According to the definition of weight vector $\mathbf{w}_{k,h}^l$ in Algorithm 2 (line 6 to line 8), we have

$$\boldsymbol{\Sigma}_{k,h}^l = \lambda \mathbf{I} + \sum_{i=1}^m \phi(s_h^{k_i}, a_h^{k_i}) \phi(s_h^{k_i}, a_h^{k_i})^\top,$$

$$\mathbf{b}_{k,h}^l = \sum_{i=1}^m \phi(s_h^{k_i}, a_h^{k_i}) \Big[ r_h(s_h^{k_i}, a_h^{k_i}) + V_{k,h+1}^l(s_{h+1}^{k_i}) \Big],$$

$$\mathbf{w}_{k,h}^l = (\mathbf{\Sigma}_{k,h}^l)^{-1}\mathbf{b}_{k,h}^l.$$

For simplicity, we omit the subscript $h$ and denote $r_{k_i} = r_h(s_h^{k_i}, a_h^{k_i}) + V_{k,h+1}^l(s_{h+1}^{k_i})$. Then for the norm $\|\mathbf{w}_k^l\|_2$, we have the following inequality

$$
\begin{aligned}
\|\mathbf{w}_k^l\|_2^2 &= \left\|(\mathbf{\Sigma}_k^l)^{-1}\sum_{i=1}^m \phi(s^{k_i}, a^{k_i})r_{k_i}\right\|_2^2 \\
&\leq m\sum_{i=1}^m \left\|(\mathbf{\Sigma}_k^l)^{-1}\phi(s^{k_i}, a^{k_i})r_{k_i}\right\|_2^2 \\
&\leq 4mH^2\sum_{i=1}^m \left\|(\mathbf{\Sigma}_k^l)^{-1}\phi(s^{k_i}, a^{k_i})\right\|_2^2 \\
&\leq \frac{4mH^2}{\lambda}\sum_{i=1}^m \phi(s^{k_i}, a^{k_i})^\top (\mathbf{\Sigma}_k^l)^{-1}\phi(s^{k_i}, a^{k_i}) \\
&= \frac{4mH^2}{\lambda}\text{tr}\left((\mathbf{\Sigma}_k^l)^{-1}\sum_{i=1}^m \phi(s^{k_i}, a^{k_i})^\top\phi(s^{k_i}, a^{k_i})\right), \quad\quad\quad \text{(E.7)}
\end{aligned}
$$

where the first inequality holds due to Cauchy-Schwarz inequality, the second inequality holds due to $|r_{k_i}| \leq 2H$ and the last inequality holds due to $\mathbf{\Sigma}_k^l \succeq \lambda I$. Now, we assume the eigen-decomposition of matrix $\sum_{i=1}^m \phi(s^{k_i}, a^{k_i})^\top\phi(s^{k_i}, a^{k_i})$ is $Q^\top\Lambda Q$ and we have

$$
\begin{aligned}
\text{tr}\left((\mathbf{\Sigma}_k^l)^{-1}\sum_{i=1}^m \phi(s^{k_i}, a^{k_i})^\top\phi(s^{k_i}, a^{k_i})\right) &= \text{tr}\left((Q^\top\Lambda Q + \lambda I)^{-1}Q^\top\Lambda Q\right) \\
&= \text{tr}\left((\Lambda + \lambda I)^{-1}\Lambda\right) \\
&= \sum_{i=1}^d \frac{\Lambda_i}{\Lambda_i + \lambda} \\
&\leq d. \quad\quad\quad\quad\quad\quad\quad \text{(E.8)}
\end{aligned}
$$

Substituting (E.8) into (E.7), we have

$$
\begin{aligned}
\|\mathbf{w}_k^l\|_2^2 &\leq \frac{4mH^2}{\lambda}\text{tr}\left((\mathbf{\Sigma}_k^l)^{-1}\sum_{i=1}^m \phi(s^{k_i}, a^{k_i})^\top\phi(s^{k_i}, a^{k_i})\right) \\
&\leq \frac{4mH^2d}{\lambda} \\
&\leq \frac{68d^2 H^3 l 4^l}{\lambda},
\end{aligned}
$$

where the first inequality holds due to (E.7), the second inequality holds due to (E.8) and the last inequality holds due to Lemma C.1. Thus, we finish the proof of Lemma C.2

$\square$

### E.3  Proof of Lemma C.3

In this section, we provide the proof of Lemma C.3. For each level $l \in \mathbb{N}$, we first denote the function class $\mathcal{V}_l$ as

$$
\begin{aligned}
\mathcal{V}_l = \bigg\{V \bigg| V(\cdot) = \max_a \min_{1 \leq i \leq l}\min\left(H, \mathbf{w}_i^\top\phi(\cdot, a) + \beta_l\sqrt{\phi(\cdot, a)^\top\mathbf{\Sigma}_i^{-1}\phi(\cdot, a)}\right), \\
\|\mathbf{w}_i\|_2 \leq 9d2^l\sqrt{H^3 l}, \mathbf{\Sigma}_i \succeq I\bigg\}. \quad\quad\quad \text{(E.9)}
\end{aligned}
$$

Therefore, for all episode $k \in K$ and stage $h \in [H]$, according to Lemma C.2, we have $\|\mathbf{w}_{k,h}^l\| \leq 9d2^l\sqrt{H^3 l}$ and it implies that the estimated value function $V_{k,h}^l \in \mathcal{V}_l$. For any function $V \in \mathcal{V}_l$, we have the following concentration property.

**Lemma E.1.** (Lemma D.4, [13]) Let $\{x_k\}_{k=1}^{\infty}$ be a real-valued stochastic process on state space $\mathcal{S}$ with corresponding filtration $\{\mathcal{F}_k\}_{k=1}^{\infty}$. Let $\{\phi_k\}_{k=1}^{\infty}$ be an $\mathbb{R}^d$-valued stochastic process where $\phi_k \in \mathcal{F}_{k-1}$ and $\|\phi_k\|_2 \leq 1$. For any $k \geq 0$, we define $\Sigma_k = I + \sum_{i=1}^{k} \phi_i \phi_i^\top$. Then with probability at least $1 - \delta$, for all $k \in \mathbb{N}$ and all function $V \in \mathcal{V}$ with $\max_s |V(x)| \leq H$, we have

$$\left\| \sum_{i=1}^{k} \phi_i \left\{ V(x_i) - \mathbb{E}\left[V(x_i)|\mathcal{F}_{i-1}\right] \right\} \right\|_{\Sigma_k^{-1}}^2 \leq 4H^2 \left[ \frac{d}{2} \log(k+1) + \log \frac{\mathcal{N}_\epsilon}{\delta} \right] + 8k^2\epsilon^2,$$

where $\mathcal{N}_\epsilon$ is the $\epsilon$-covering number of the function class $\mathcal{V}$ with respect to the distance function $\text{dist}(V_1, V_2) = \max_s |V_1(s) - V_2(s)|$.

Furthermore, for each function class $\mathcal{V}_l$, the covering number $\mathcal{N}_\epsilon$ of $\mathcal{V}_l$ can be upper bounded by following Lemma.

**Lemma E.2.** For each function class $\mathcal{V}_l$, we define the distance between two function $V_1$ and $V_2$ as $V_1, V_2 \in \mathcal{V}_l$ as $dist(V_1, V_2) = \max_s |V_1(s) - V_2(s)|$. With respect to this distance function, the $\epsilon$-covering number $\mathcal{N}_\epsilon$ of the function class $\mathcal{V}_l$ can be upper bounded by

$$\log \mathcal{N}_\epsilon \leq dl \log(1 + 36d2^l \sqrt{H^3 l}/\epsilon) + d^2 l \log(1 + 8\sqrt{d}\beta_l^2/\epsilon^2).$$

*Proof.* See Appendix F.1. $\qquad\square$

*Proof of Lemma C.3.* Similar to the proof of Lemma B.2, we first focus on a fixed level $l \in \mathbb{N}$ and a fixed stage $h \in [H]$. Now, we denote $k_0 = 0$, and for $i \in \mathbb{N}$, we denote $k_i$ as the minimum index of the episode where the action is added to the set $\mathcal{C}_h^l$:

$$k_i = \min\left\{ k : k > k_{i-1}, l_h^k = l \right\}. \tag{E.10}$$

Now, we consider the $\sigma$-algebra filtration $\mathcal{F}_i = \sigma(s_1^1, ..., s_H^1, s_1^2, ..., s_H^2, .., s_1^{k_{i+1}}, .., s_h^{k_{i+1}})$ that contain all randomness before receive the reward $r_h(s_h^{k_{i+1}}, a_h^{k_{i+1}})$ and next state $s_{h+1}^{k_{i+1}}$ at episode $k_{i+1}$. By this definition, $\phi(s_h^{k_i}, a_h^{k_i})$ is $\mathcal{F}_{i-1}$-measurable and the next state $s_{h+1}^{k_i}$ is $\mathcal{F}_i$-measurable. Since the randomness in this filtration only comes from the stochastic state transition process $s_{h+1} \sim \mathbb{P}_h(\cdot|s_h, a_h)$ and we determine the level $l_h^k$ before receive the reward $r_h(s_h^k, a_h^k)$ and next state $s_{h+1}^k$ at episode $k$, for any fixed value function $V \in \mathcal{V}_l$, we have

$$\mathbb{E}\left[V(s_{h+1}^{k_i})|\mathcal{F}_{i-1}\right] = [\mathbb{P}_h V](s_h^{k_i}, a_h^{k_i}). \tag{E.11}$$

According to Lemma E.1 with probability at least $1 - \delta/\big(H 2 l(l+1)\big)$, for all number $i \in \mathbb{N}$ and all function $V \in \mathcal{V}_l$, we have

$$\left\| \sum_{j=1}^{i} \phi(s_h^{k_j}, a_h^{k_j}) \left[ V(s_{h+1}^{k_j}) - [\mathbb{P}_h V](s_h^{k_j}, a_h^{k_j}) \right] \right\|_{(\Sigma_{k_{i+1},h}^l)^{-1}}$$

$$= \left\| \sum_{j=1}^{i} \phi(s_h^{k_j}, a_h^{k_j}) \left[ V(s_{h+1}^{k_j}) - \mathbb{E}\left[V(s_{h+1}^{k_{i+1}})|\mathcal{F}_{i-1}\right] \right] \right\|_{(\Sigma_{k_{i+1},h}^l)^{-1}}$$

$$\leq 4H^2 \left[ \frac{d}{2} \log(i+1) + \log \frac{\mathcal{N}_\epsilon}{\delta/\big(H 2 l(l+1)\big)} \right] + 8i^2\epsilon^2$$

$$\leq 4H^2 \left[ \frac{d}{2} \log(i+1) + dl \log(1 + 36d2^l \sqrt{H^3 l}/\epsilon) + d^2 l \log(1 + 8\sqrt{d}\beta_l^2/\epsilon^2) \right.$$

$$\left. + \log(2Hl(l+1)/\delta) \right] + 8i^2\epsilon^2, \tag{E.12}$$

where the first inequality holds due to Lemma E.1 and the second inequality holds due to Lemma E.2. Furthermore, Lemma C.1 show that the size of set $|\mathcal{C}_h^l|$ is upper bounded by $|\mathcal{C}_h^l| \leq 17 dl H 4^l$. This reuslt implies that $k_{17 dl H 4^l + 1} = \infty$ and we only need to consider episode $k_i$ for $i \leq 17 dl H 4^l + 1$.

Now, we choose $\epsilon = 1/(17l4^l)$, then for all episode $k_i < k \le k_{i+1}$ and function $V = V_{k,h+1}^l \in \mathcal{V}_l$, we have

$$\left\| \sum_{j=1}^i \phi(s_h^{k_j}, a_h^{k_j}) \big[ V_{k,h+1}^l(s_{h+1}^{k_j}) - [\mathbb{P}_h V_{k,h+1}^l](s_h^{k_j}, a_h^{k_j}) \big] \right\|_{(\boldsymbol{\Sigma}_{k,h}^l)^{-1}}^2$$

$$\le 4H^2 \Bigg[ \frac{dl}{2} \log(69dlH) + 2dl^2 \log(1 + 2448d\sqrt{H^3l^3})$$

$$+ d^2l^2 \log(1 + 36992\sqrt{d}l^2\beta_l^2) + 2\log(4lH/\delta) \Bigg] + 8d^2H^2, \tag{E.13}$$

where the first inequality holds due to (E.12) with the fact that $\boldsymbol{\Sigma}_{k,h}^l$ does not change for $k_i < k \le k_{i+1}$ and $i \le 17dlH4^l + 1$. Finally, taking an union bound for all level $l \in \mathbb{N}$ and all stage $h \in [H]$, with probability at $1 - \delta/2$, for all level $l \in \mathbb{N}$, all stage $h \in [H]$ and all episode $k \in \mathbb{N}$, we have

$$\left\| \sum_{i \in \mathcal{C}_{k,h}^l} \phi(s_h^i, a_h^i) \big[ V_{k,h+1}^l(s_{h+1}^i) - [\mathbb{P}_h V_{k,h+1}^l](s_h^i, a_h^i) \big] \right\|_{(\boldsymbol{\Sigma}_{k,h}^l)^{-1}}^2 \le Cd^2 H^2 l^2 \log(dlH\beta_l^2/\delta),$$

where $C$ is a large absolute constant. Thus, we finish the proof of Lemma C.3. $\qquad\square$

## E.4 Proof of Lemma C.4

**Lemma E.3.** [Lemma B.1, [13]] Under Assumption 3.1, for any fixed policy $\pi$, there exists a series of vectors $\{\mathbf{w}_h^\pi\}_{h=1}^H$, such that for all state-action pair $(s,a) \in \mathcal{S} \times \mathcal{A}$ and all stage $h \in [H]$, we have

$$Q_h^\pi(s,a) = (\mathbf{w}_h^\pi)^\top \phi(s,a), \|\mathbf{w}_h^\pi\| \le 2H\sqrt{d}.$$

*Proof of Lemma C.4.* For simplicity, let $\mathcal{C}_{k,h}^l = \{k_1, .., k_m\}$ denote the index set $\mathcal{C}_h^l$ at the beginning of episode $k$, where $1 \le k_1 < k_2 < .. < k_m < k$. According to Lemma E.3, for each fixed policy $\pi$, there exists a vector $\mathbf{w}_h^\pi$ such that

$$(\mathbf{w}_h^\pi)^\top \phi(s,a) = Q_h^\pi(s,a) = r_h(s,a) + \big[\mathbb{P}_h V_{h+1}^\pi\big](s,a). \tag{E.14}$$

According to the definition of vector $\mathbf{w}_{k,h}$ in Algorithm 2 (line 6 to line 8), we have

$$\boldsymbol{\Sigma}_{k,h}^l = \lambda \mathbf{I} + \sum_{i=1}^m \phi(s_h^{k_i}, a_h^{k_i}) \phi(s_h^{k_i}, a_h^{k_i})^\top,$$

$$\mathbf{b}_{k,h}^l = \sum_{i=1}^m \phi(s_h^{k_i}, a_h^{k_i}) \Big[ r_h(s_h^{k_i}, a_h^{k_i}) + V_{k,h+1}^l(s_{h+1}^{k_i}) \Big],$$

$$\mathbf{w}_{k,h}^l = (\boldsymbol{\Sigma}_{k,h}^l)^{-1} \mathbf{b}_{k,h}^l. \tag{E.15}$$

For simplicity, we omit the subscript $l$ and combining (E.14) and (E.15), we have

$$\mathbf{w}_{k,h} - \mathbf{w}_h^\pi = \boldsymbol{\Sigma}_{k,h}^{-1} \sum_{i=1}^m \phi(s_h^{k_i}, a_h^{k_i}) \big[ r_h(s_h^{k_i}, a_h^{k_i}) + V_{k,h+1}(s_{h+1}^{k_i}) \big] - \mathbf{w}_h^\pi$$

$$= \boldsymbol{\Sigma}_{k,h}^{-1} \bigg[ -\lambda \mathbf{w}_h^\pi - \sum_{i=1}^m \phi(s_h^{k_i}, a_h^{k_i}) \phi(s_h^{k_i}, a_h^{k_i})^\top \mathbf{w}_h^\pi$$

$$+ \sum_{i=1}^m \phi(s_h^{k_i}, a_h^{k_i}) \big[ r_h(s_h^{k_i}, a_h^{k_i}) + V_{k,h+1}(s_{h+1}^{k_i}) \big] \bigg]$$

$$= \boldsymbol{\Sigma}_{k,h}^{-1} \bigg[ -\lambda \mathbf{w}_h^\pi + \sum_{i=1}^m \phi(s_h^{k_i}, a_h^{k_i}) \Big( V_{k,h+1}(s_{h+1}^{k_i}) - [\mathbb{P}_h V_{h+1}^\pi](s_h^{k_i}, a_h^{k_i}) \Big) \bigg]$$

$$= \underbrace{-\lambda \mathbf{\Sigma}_{k,h}^{-1} \mathbf{w}_h^\pi}_{I_1} + \underbrace{\mathbf{\Sigma}_{k,h}^{-1} \sum_{i=1}^m \phi(s_h^{k_i}, a_h^{k_i}) \Big( V_{k,h+1}(s_{h+1}^{k_i}) - [\mathbb{P}_h V_{k,h+1}](s_h^{k_i}, a_h^{k_i}) \Big)}_{I_2}$$

$$+ \underbrace{\mathbf{\Sigma}_{k,h}^{-1} \sum_{i=1}^m \phi(s_h^{k_i}, a_h^{k_i}) \big[ \mathbb{P}_h (V_{k,h+1} - V_{h+1}^\pi) \big](s_h^{k_i}, a_h^{k_i})}_{I_3},$$

where the third equality holds due to (E.14). For the term $I_1$ and any state-action pair $(s,a) \in \mathcal{S} \times \mathcal{A}$, we have

$$\begin{aligned}
\big| \langle I_1, \phi(s,a) \rangle \big| &= \big| \lambda \phi(s,a)^\top \mathbf{\Sigma}_{k,h}^{-1} \mathbf{w}_h^\pi \big| \\
&\leq \lambda \big\| \phi(s,a)^\top \mathbf{\Sigma}_{k,h}^{-1} \big\|_2 \| \mathbf{w}_h^\pi \|_2 \\
&\leq \sqrt{\lambda} \| \mathbf{w}_h^\pi \|_2 \sqrt{\phi(s,a)^\top \mathbf{\Sigma}_{k,h}^{-1} \phi(s,a)} \\
&\leq 2H\sqrt{d\lambda} \sqrt{\phi(s,a)^\top \mathbf{\Sigma}_{k,h}^{-1} \phi(s,a)},
\end{aligned} \tag{E.16}$$

where the first inequality holds due to Cauchy-Schwarz inequality, the second inequality holds due to $\mathbf{\Sigma}_{k,h} \succeq \lambda I$ and the third inequality holds due to Lemma E.3. For the term $I_2$ and any state-action pair $(s,a) \in \mathcal{S} \times \mathcal{A}$, according to Lemma C.3, we have

$$\begin{aligned}
\big| \langle I_2, \phi(s,a) \rangle \big| &\leq \sqrt{\phi(s,a)^\top \mathbf{\Sigma}_{k,h}^{-1} \phi(s,a)} \\
&\quad \cdot \left\| \sum_{i=1}^m \phi(s_h^{k_i}, a_h^{k_i}) \big[ V_{k,h+1}(s_{h+1}^{k_i}) - [\mathbb{P}_h V_{k,h+1}](s_h^{k_i}, a_h^{k_i}) \big] \right\|_{\mathbf{\Sigma}_{k,h}^{-1}} \\
&\leq CdHl\sqrt{\log(dlH\beta_l^2/\delta)} \sqrt{\phi(s,a)^\top \mathbf{\Sigma}_{k,h}^{-1} \phi(s,a)},
\end{aligned} \tag{E.17}$$

where the first inequality holds due to Cauchy-Schwarz inequality and the second inequality holds due to Lemma C.3. For the term $I_3$ and any state-action pair $(s,a) \in \mathcal{S} \times \mathcal{A}$, we have

$$\begin{aligned}
\langle \phi(s,a), I_3 \rangle &= \left\langle \phi(s,a), \mathbf{\Sigma}_{k,h}^{-1} \sum_{i=1}^m \phi(s_h^{k_i}, a_h^{k_i}) \big[ \mathbb{P}_h(V_{k,h+1} - V_{h+1}^\pi) \big](s_h^{k_i}, a_h^{k_i}) \right\rangle \\
&= \left\langle \phi(s,a), \mathbf{\Sigma}_{k,h}^{-1} \sum_{i=1}^m \phi(s_h^{k_i}, a_h^{k_i}) \phi(s_h^{k_i}, a_h^{k_i})^\top \int (V_{k,h+1} - V_{h+1}^\pi)(s') d\boldsymbol{\mu}_h(s') \right\rangle \\
&= \underbrace{\left\langle \phi(s,a), \int (V_{k,h+1} - V_{h+1}^\pi)(s') d\boldsymbol{\mu}_h(s') \right\rangle}_{J_1} \\
&\quad - \lambda \underbrace{\left\langle \phi(s,a), \mathbf{\Sigma}_{k,h}^{-1} \int (V_{k,h+1} - V_{h+1}^\pi)(s') d\boldsymbol{\mu}_h(s') \right\rangle}_{J_2},
\end{aligned}$$

For term $J_1$, we have

$$\begin{aligned}
J_1 &= \left\langle \phi(s,a), \int (V_{k,h+1} - V_{h+1}^\pi)(s') d\boldsymbol{\mu}_h(s') \right\rangle \\
&= \int \left\langle \phi(s,a), (V_{k,h+1} - V_{h+1}^\pi)(s') \right\rangle d\boldsymbol{\mu}_h(s') \\
&= \int \mathbb{P}_h(s'|s,a)(V_{k,h+1} - V_{h+1}^\pi)(s') \rangle ds' \\
&= \big[ \mathbb{P}_h(V_{k,h+1} - V_{h+1}^\pi) \big](s,a).
\end{aligned} \tag{E.18}$$

For term $J_2$, we have

$$\big| J_2 \big| = \lambda \left| \left\langle \phi(s,a), \mathbf{\Sigma}_{k,h}^{-1} \int (V_{k,h+1} - V_{h+1}^\pi)(s') d\boldsymbol{\mu}_h(s') \right\rangle \right|$$

$$\leq \lambda \big\| \phi(s,a)^\top \mathbf{\Sigma}_{k,h}^{-1} \big\|_2 \left\| \int (V_{k,h+1} - V_{h+1}^\pi)(s') d\boldsymbol{\mu}_h(s') \right\|_2$$

$$\leq \sqrt{d}\lambda \big\| \phi(s,a)^\top \mathbf{\Sigma}_{k,h}^{-1} \big\|_2 \max_{s'} \big| (V_{k,h+1} - V_{h+1}^\pi)(s') \big|$$

$$\leq 2H\sqrt{d}\lambda \big\| \phi(s,a)^\top \mathbf{\Sigma}_{k,h}^{-1} \big\|_2$$

$$\leq 2H\sqrt{d}\lambda \sqrt{\phi(s,a)^\top \mathbf{\Sigma}_{k,h}^{-1} \phi(s,a)}, \tag{E.19}$$

where the first inequality holds due to Cauchy-Schwarz inequality, the second inequality holds due to Assumption 3.1, the third inequality holds because of $\big| (V_{k,h+1} - V_{h+1}^\pi)(s') \big| \leq 2H$ and the last inequality holds due to $\mathbf{\Sigma}_{k,h} \succeq \lambda I$. Combining (E.16),(E.17),(E.18),(E.19) with the fact that $\lambda = 1$, we have

$$\Big| \langle \phi(s,a), \mathbf{w}_{k,h} \rangle - Q_h^\pi(s,a) - \big[ \mathbb{P}_h(V_{k,h+1} - V_{h+1}^\pi) \big](s,a) \Big|$$

$$= |J_2 + \langle I_1, \phi(s,a) \rangle + \langle I_2, \phi(s,a) \rangle|$$

$$\leq \left( CdHl\sqrt{\log(dlH\beta_l^2/\delta)} + 4H\sqrt{d} \right) \sqrt{\phi(s,a)^\top (\mathbf{\Sigma}_{k,h})^{-1} \phi(s,a)}.$$

Notice that there exists a large constant $C'$ such that for all level $l \in \mathbb{N}$ with parameter $\beta_l = C'dHl\sqrt{\log(dlH/\delta)}$, the following inequality hods:

$$CdHl\sqrt{\log(dlH\beta_l^2/\delta)} + 4H\sqrt{d} \leq C'dHl\sqrt{\log(dlH/\delta)}. \tag{E.20}$$

When (E.20) holds, we further have

$$\Big| \langle \phi(s,a), \mathbf{w}_{k,t} \rangle - Q_h^\pi(s,a) - \big[ \mathbb{P}_h(V_{k,h+1} - V_{h+1}^\pi) \big](s,a) \Big|$$

$$\leq \left( CdHl\sqrt{\log(dlH\beta_l^2/\delta)} + 3H\sqrt{d} \right) \sqrt{\phi(s,a)^\top (\mathbf{\Sigma}_{k,h})^{-1} \phi(s,a)}$$

$$\leq C'dHl\sqrt{\log(dlH/\delta)} \sqrt{\phi(s,a)^\top (\mathbf{\Sigma}_{k,h})^{-1} \phi(s,a)}$$

$$= \beta_l \sqrt{\phi(s,a)^\top (\mathbf{\Sigma}_{k,h})^{-1} \phi(s,a)}.$$

Thus, we finish the proof of Lemma C.4. □

## E.5 Proof of Lemma C.5

*Proof of Lemma C.5.* Now, we use induction to prove this lemma. First, we prove the base case. For all state $s \in \mathcal{S}$ and level $l \in \mathbb{N}$, we have $V_{k,H+1}^l(s) = 0 = V_{H+1}^*(s)$. Second, if $V_{k,h+1}^l(s) \geq V_{h+1}^*(s)$ holds for all state $s \in \mathcal{S}$ and level $l \in \mathbb{N}$ at stage $h+1$, then for any state $s \in \mathcal{S}$ and level $l \in \mathbb{N}$ at stage $h$, we have

$$(\mathbf{w}_{k,h}^l)^\top \phi(s,a) + \beta_l \sqrt{\phi(s,a)^\top (\Sigma_{k,h}^l)^{-1} \phi(s,a)} - Q_h^*(s,a) \geq \big[ \mathbb{P}_h(V_{k,h+1}^l - V_{h+1}^*) \big](s,a) \geq 0,$$

where the first inequality holds due to Lemma C.4 and the second inequality holds due to the induction assumption. Furthermore, the optimal value function is upper bounded by $Q_h^*(s,a) \leq H$ and it implies that

$$Q_h^*(s,a) \leq \min \left( (\mathbf{w}_{k,h}^l)^\top \phi(s,a) + \beta_l \sqrt{\phi(s,a)^\top (\Sigma_{k,h}^l)^{-1} \phi(s,a)}, H \right) = Q_{k,h}^l(s,a). \tag{E.21}$$

Thus, for each level $l \in \mathbb{N}$ and state $s \in \mathcal{S}$, we have

$$V_{k,h}^l(s) = \max_a \min_{1 \leq i \leq l} Q_{k,h}^i(s,a)$$

$$\geq \max_a \min_{1 \leq i \leq l} Q_h^*(s,a)$$

$$= \max_a Q_h^*(s,a)$$

$$= V_h^*(s),$$

where the inequality holds due to (E.21). Finally, by induction, we finish the proof of Lemma C.5. □

# F    Auxiliary Lemmas

**Lemma F.1** (Azuma–Hoeffding inequality, [4]). Let $\{x_i\}_{i=1}^n$ be a martingale difference sequence with respect to a filtration $\{\mathcal{G}_i\}$ satisfying $|x_i| \leq M$ for some constant $M$, $x_i$ is $\mathcal{G}_{i+1}$-measurable, $\mathbb{E}[x_i|\mathcal{G}_i] = 0$. Then for any $0 < \delta < 1$, with probability at least $1 - \delta$, we have

$$\sum_{i=1}^n x_i \leq M\sqrt{2n\log(1/\delta)}.$$

## F.1    Proof of Lemma E.2

We need the following Lemma:

**Lemma F.2** (Lemma D.5, [13]). For an Euclidean ball with radius $R$ in $\mathbb{R}^d$, the $\epsilon$-covering number of this ball is upper bounded by $(1 + 2R/\epsilon)^d$.

*Proof of Lemma E.2.* For any two function $V_1, V_2 \in \mathcal{V}_l$, according to the definition of function class $\mathcal{V}_l$, we have

$$V_1(\cdot) = \max_a \min_{1\leq i\leq l} \min\left(H, \mathbf{w}_{1,i}^\top\boldsymbol{\phi}(\cdot, a) + \beta_l\sqrt{\boldsymbol{\phi}(\cdot, a)^\top\boldsymbol{\Gamma}_{1,i}\boldsymbol{\phi}(\cdot, a)}\right),$$

$$V_2(\cdot) = \max_a \min_{1\leq i\leq l} \min\left(H, \mathbf{w}_{2,i}^\top\boldsymbol{\phi}(\cdot, a) + \beta_l\sqrt{\boldsymbol{\phi}(\cdot, a)^\top\boldsymbol{\Gamma}_{2,i}\boldsymbol{\phi}(\cdot, a)}\right),$$

where $\|\mathbf{w}_{1,i}\|_2, \|\mathbf{w}_{2,i}\|_2 \leq 9d2^l\sqrt{H^3l}$ and $\boldsymbol{\Gamma}_{1,i}, \boldsymbol{\Gamma}_{2,i} \preceq \mathbf{I}$. Since all of the functions $\max_a$, $\min_{1\leq i\leq l}$ and $\min(H, \cdot)$ are contraction functions, we have

$$\begin{aligned}
\text{dist}(V_1, V_2) &= \max_{s\in\mathcal{S}}\left|V_1(s) - V_2(s)\right| \\
&\leq \max_{1\leq i\leq l, s\in\mathcal{S}, a\in\mathcal{A}}\left|\mathbf{w}_{1,i}^\top\boldsymbol{\phi}(s,a) + \beta_l\sqrt{\boldsymbol{\phi}(s,a)^\top\boldsymbol{\Gamma}_{1,i}\boldsymbol{\phi}(s,a)}\right. \\
&\qquad\qquad \left. - \mathbf{w}_{2,i}^\top\boldsymbol{\phi}(s,a) - \beta_l\sqrt{\boldsymbol{\phi}(s,a)^\top\boldsymbol{\Gamma}_{2,i}\boldsymbol{\phi}(s,a)}\right| \\
&\leq \beta_l\max_{1\leq i\leq l, s\in\mathcal{S}, a\in\mathcal{A}}\left|\sqrt{\boldsymbol{\phi}(s,a)^\top\boldsymbol{\Gamma}_{1,i}\boldsymbol{\phi}(s,a)} - \sqrt{\boldsymbol{\phi}(s,a)^\top\boldsymbol{\Gamma}_{2,i}\boldsymbol{\phi}(s,a)}\right| \\
&\qquad + \max_{1\leq i\leq l, s\in\mathcal{S}, a\in\mathcal{A}}\left|(\mathbf{w}_{1,i} - \mathbf{w}_{2,i})^\top\boldsymbol{\phi}(s,a)\right| \\
&\leq \beta_l\max_{1\leq i\leq l, s\in\mathcal{S}, a\in\mathcal{A}}\left|\sqrt{\boldsymbol{\phi}(s,a)^\top(\boldsymbol{\Gamma}_{1,i} - \boldsymbol{\Gamma}_{2,i})\boldsymbol{\phi}(s,a)}\right| \\
&\qquad + \max_{1\leq i\leq l, s\in\mathcal{S}, a\in\mathcal{A}}\left|(\mathbf{w}_{1,i} - \mathbf{w}_{2,i})^\top\boldsymbol{\phi}(s,a)\right| \\
&\leq \beta_l\max_{1\leq i\leq l}\sqrt{\|\boldsymbol{\Gamma}_{1,i} - \boldsymbol{\Gamma}_{2,i}\|_F} + \max_{1\leq i\leq l}\|\mathbf{w}_{1,i} - \mathbf{w}_{2,i}\|_2, \quad\quad\quad\text{(F.1)}
\end{aligned}$$

where the first inequality holds due to the contraction property, the second inequality holds due to the fact that $\max_x|f(x) + g(x)| \leq \max_x|f(x)| + \max_x|g(x)|$, the third inequality holds due to $|\sqrt{x} - \sqrt{y}| \geq |\sqrt{x} - \sqrt{y}|$ and the last inequality holds due to the fact that $\|\boldsymbol{\phi}(s,a)\|_2 \leq 1$. Now, we denote $\mathcal{C}_{\mathbf{w}}$ as a $\epsilon/2$-cover of the set $\left\{\mathbf{w}\in\mathbb{R}^d\,\big|\,\|\mathbf{w}\|_2 \leq 9d2^l\sqrt{H^3l}\right\}$ and $\mathcal{C}_{\boldsymbol{\Gamma}}$ as a $\epsilon^2/(4\beta_l^2)$-cover of the set $\left\{\boldsymbol{\Gamma}\in\mathbb{R}^{d\times d}\,\big|\,\|\boldsymbol{\Gamma}\|_F \leq \sqrt{d}\right\}$ with respect to the Frobenius norms. Thus, according to Lemma F.2, we have following property:

$$|\mathcal{C}_{\mathbf{w}}| \leq \left(1 + 36d2^l\sqrt{H^3l}/\epsilon\right)^d, |\mathcal{C}_{\boldsymbol{\Gamma}}| \leq \left(1 + 8\sqrt{d}\beta_l^2/\epsilon^2\right)^{d^2}. \quad\quad\quad\text{(F.2)}$$

By the definition of covering number, for any function $V_1 \in \mathcal{V}_l$ with parameters $\mathbf{w}_{1,i}, \boldsymbol{\Gamma}_{1,i}(1 \leq i \leq l)$, there exists other parameters $\mathbf{w}_{2,i}, \boldsymbol{\Gamma}_{2,i}(1 \leq i \leq l)$ such that $\mathbf{w}_{2,i} \in \mathcal{C}_{\mathbf{w}}, \boldsymbol{\Gamma}_{2,i} \in \mathcal{C}_{\boldsymbol{\Gamma}}$ and $\|\mathbf{w}_{2,i} - \mathbf{w}_{1,i}\|_2 \leq \epsilon/2, \|\boldsymbol{\Gamma}_{2,i} - \boldsymbol{\Gamma}_{1,i}\|_F \leq \epsilon^2/(4\beta_l^2)$. Thus, we have

$$\text{dist}(V_1, V_2) \leq \beta_l\max_{1\leq i\leq l}\sqrt{\|\boldsymbol{\Gamma}_{1,i} - \boldsymbol{\Gamma}_{2,i}\|_F} + \max_{1\leq i\leq l}\|\mathbf{w}_{1,i} - \mathbf{w}_{2,i}\|_2 \leq \epsilon,$$

where the inequality holds due to (F.1). Therefore, the $\epsilon$-covering number of function class $\mathcal{V}_l$ is bounded by $\mathcal{N}_\epsilon \leq |\mathcal{C}_{\mathbf{w}}|^l \cdot |\mathcal{C}_{\boldsymbol{\Gamma}}|^l$ and it implies

$$\log \mathcal{N}_\epsilon \leq dl \log(1 + 36d2^l \sqrt{H^3 l}/\epsilon) + d^2 l \log(1 + 8\sqrt{d}\beta_l^2/\epsilon^2),$$

where the first inequality holds due to (F.2). Thus, we finish the proof of Lemma E.2. $\qquad\square$