# OpenReview forum: "Uniform-PAC Bounds for Reinforcement Learning with Linear Function Approximation"
_NeurIPS.cc/2021/Conference — NeurIPS 2021 Poster_

### Official Review · Reviewer_Bmer · 2021-07-16

**Rating:** 6
**Confidence:** 3

**Summary:**

This paper designs an algorithm with uniform-PAC for reinforcement learning with linear function approximation.

**Limitations And Societal Impact:**

This paper focuses on theoretical analysis. It's expected that the paper does not have direct negative societal impact.

**Main Review:**

This is the first uniform-PAC algorithm for RL beyond tabular setting. This paper also proves sample complexity on the same order of that of algorithms which are not uniform-PAC.

A major question is in Section 3, the action space $\mathcal{A}$ is required to be finite. Is this requirement fundamental to the design of the algorithms? Can this condition be relaxed?

**Time Spent Reviewing:**

1.5

---

> ### Author Response · Authors · 2021-08-09
> **Response to Reviewer Bmer**
>
> Q1: The action space $A$ is required to be finite. Is it fundamental? Can it be relaxed?
>
> A1: This finite action space assumption is not fundamental to the design and analysis of the algorithm.  The only reason we make this assumption is that in order to compute the value function in line 12 of Algorithm 2, and compute the greedy policy in line 17 of Algorithm 2, finite action space will ensure the algorithm to be computationally efficient. Such an assumption has also been made (implicitly or explicitly) in almost all prior works on reinforcement learning with linear function approximation (e.g., [3,10,13,17,27]). This assumption can be relaxed if we assume the maximization over the infinite action space can be done efficiently, or if one does not care about the computational cost of this part.

---

### Official Review · Reviewer_4pTr · 2021-07-18

**Rating:** 7
**Confidence:** 5

**Summary:**

The paper a proposes optimistic algorithm for reinforcement learning in linear MDPs and proves the first uniform-PAC bound for this setting. Unlike existing result, this stronger notion of a PAC bound ensures high-probability regret as well as convergence to the optimal policy. Along the way, the paper also provides an algorithm with uniform-PAC guarantees for the contextual linear bandit setting. The algorithms are similar to existing UCB algorithms but employ a sample-splitting technique. This technique essentially ensures the following key property for uniform-PAC results: the confidence bound for a certain state-action pair cannot increase arbitrarily with the number of episodes (as is the case with existing UCBs that typically have log(T) dependence in this setting and therefore increase with T unless an informative observation was gathered).


**Limitations And Societal Impact:**

The paper has adequately addressed limitations and potential negative societal impact.

**Main Review:**

This is a solid paper with signifiant and technically interesting contributions.

Significance: Algorithms that can be run anytime and are guarantees to converge to the optimal policy are highly desirable since in practical scenarios, one is often interested in extracting a single policy eventually but does not know the number of episodes (desired accuracy level epsilon) ahead of time. Existing online-to-batch solutions (e.g. those mentioned for UCB-LSVI) that convert regret to PAC-style guarantees are based on randomly picking one of the policies executed by the algorithm but this is limited to PAC guarantees with 1/delta dependence instead of log(1/delta) and are thus too loose when the failure probability delta is small.
This work is the first to provide stronger guarantees that imply regret and PAC bounds simultaneously for all epsilon-levels in the function approximation setting, based on the notion of uniform-PAC that has been proposed for the tabular setting. This is an important result and the paper provides technical tools that will be of interest in extending such guarantees beyond the linear setting considered here.

Originality: Using sample splitting in bandits and RL is not new. It has been used for variance reduction in tabular RL as well as earlier algorithms (cf. MoRMax or Delayed-Q learning). However, to the best of my knowledge, the kind of splitting and analysis in this paper is novel, as well as the use to achieve uniform-PAC bounds. I expect this to be a useful tool for other settings as well.

Quality: I have checked the main novel components of the analysis (e.g. Lemma A1 / B1 and A2 / B2 as well as the proof of the main theorems. Besides minor typos, I did not encounter any issues. The analysis is overall well done and presented in a sufficiently detailed and clear way.

Presentation: Overall, the paper does a good job at conveying the main intuition and the details of the analysis. That said, I encourage the authors to do a careful pass on the paper for typos and to improve the writing in certain places (see detailed comments below).

Detailed comments and typos:
- Line 54: round -> rounds
- Line 59: the RL -> RL
- Line 59: on the -> in
- Section 2.1: the writing is extremely monotonous and should be substantially improved. Every sentence has the exact same structure. It would also be more helpful to relate the existing works better to the contributions made in this paper.
- Line 141: suboptimality for -> suboptimality gap for
- Remark 3.5: I recommend mentioning that sampling a policy uniformly from the policies executed by the algorithm leads to a poly(1/delta) sample complexity result (and is therefore not considered PAC in the strict original definition) as opposed to a log(1/delta) achieved by the uniform-PAC bound presented in this submission.
- Line 179: Why not just use general sub-Gaussian noise instead of bounding the magnitude? Also, why did the authors adopt a stronger l2 bound on the parameters for the bandit setting (1 vs sqrt(d))? This seems not necessary?
- Sec. 5: provide a reference to Alg 2 in the text.
- Factor H difference compared to UCB-LSVI: The additional factor of sqrt(H) in the regret bound seems to come from the bound in the number of samples per bucket. From what I understand, this is due to the fact that the level l at h can be at most the level at h-1 per episode and therefore additional elements may be added to the levels even though the first condition in line 19 of Alg 2 would hold for higher levels l. This makes the recursive bound on the gap go through nicely but I wonder whether one could avoid that and allow the levels to increase with h per episode (and thus save the additional H dependency). Did the authors explore this direction?
- Line 278: , we -> . We
- Line 300: pair -> pairs
- Line 304 level satisfies -> level that satisfies
- The paper does not discuss the computational cost of the algorithm. By the design of the levels, I think one can show that the number of level can only be logarithmic in k which is not a large additional computational burden. I recommend adding this result and a discussion.

**Time Spent Reviewing:**

3

---

> ### Author Response · Authors · 2021-08-09
> **Response to Reviewer 4pTr**
>
> Thank you for your positive feedback
>
> Q1: Writing suggestion and some existed typos
>
> A1: We apologize for the typos. We will fix them in the final version and improve the presentation in those several places.
>
> Q2: Why not just use general sub-Gaussian noise instead of bounding the magnitude? Also, why did the authors adopt a stronger $l_2$ bound on the parameters for the bandit setting (1 vs. $\sqrt{d}$)? This seems not necessary?
>
> A2: Thanks for your suggestion. Actually, our result in linear bandits can be easily extended to the sub-Gaussian case with the same algorithm and almost the same analysis (i.e., all the concentration inequalities used in our analysis are directly applicable to sub-Gaussian random variables). It is very easy to make this change, and the revision is very minor.
>
> We assume the $\ell_2$ norm of the parameter vector in linear bandits to be bounded by 1,  just in order to make it consistent with the assumption in the OFUL paper [1]. Our assumption can be relaxed to be bounded by $\sqrt{d}$ very easily. We will emphasize this point in the final version.
>
> Q3: The additional factor of $\sqrt{H}$ and whether one could avoid that and allow the levels to increase with h per episode?
>
> A3: You are right that the additional factor of $\sqrt{H}$ comes from the monotonic property of the levels across different stages h. We need the monotonic property to prove inequality (B.5) (Appendix, line 488). We have tried to remove this constraint. But did not find a way to get rid of it yet. We will continue to explore this direction in the future.
>
> Q4: Computation cost
>
> A4: You are right that the multi-layer structure will only increase the computation cost by a factor of $\log K$. We will add a discussion to emphasize this fact.

---

> > ### Comment · Reviewer_4pTr · 2021-08-22
> > **Re: Response to Reviewer 4pTr**
> >
> > Thank you for your response. I think this is a good paper and am confident the authors can make the necessary minor changes. I maintain my score and vote for acceptance.

---

### Official Review · Reviewer_FX5f · 2021-07-22

**Rating:** 7
**Confidence:** 4

**Summary:**

In this paper, the authors develop a new algorithm for linear bandits that achieves a uniform-PAC bound of $\tilde{O}(d^2/\epsilon^2)$, meaning that for all $\epsilon > 0$, the algorithm outputs an action whose suboptimality is at most $\epsilon$ after $\tilde{O}(d^2/\epsilon^2)$ rounds of iterations. The authors further generalize their techniques to linear MDPs and develop an algorithm that achieves a uniform-PAC bound of $\tilde{O}(d^3H^5/\epsilon^2)$.

**Limitations And Societal Impact:**

See main review.

**Main Review:**

Uniform-PAC is a notion developed by Dann et al. [8], which says that with probability at least $1-\delta$, for any $\epsilon > 0$, the algorithm always outputs an $\epsilon$-optimal policy after $f(\epsilon, \delta)$ rounds of iterations. It is known that such guarantee is stronger than the usual PAC guarantee and regret guarantee. In this paper, the authors develop uniform-PAC algorithms for linear bandits and RL with linear function approximation.

The main technical contribution of this paper is a new trick for developing UCB-based algorithms with uniform-PAC guarantee in the linear setting. The authors argue that existing algorithms may not satisfy the uniform-PAC guarantee either because they need to know the number of rounds of iterations in advance, or their confidence bound has a $\log k$ factor which makes it possible for the algorithm to output suboptimal policies infinitely often. To remove the $\log k$ factor in existing UCB-based algorithms, inspired by [6], the authors develop a clever layering method. Specifically, the authors classify all existing actions/data points into groups according to their uncertainty bonus, and each action/data point is added to a group $k$ so that the uncertainty bonus is at most $2^{-k}$. This makes sure that the size of group $k$ is at most $\mathrm{exp}(k)$. For the analysis, instead of doing a union bound over all existing data points which will introduce a $\log k$ factor in the confidence bound, for this new algorithm, we set the failure probably of group $k$ to be roughly $2^{-k}$. Doing so will introduce a factor of $k$ in the confidence bound of group $k$, but that will be ok for the algorithm because the uncertainty bonus is at most $2^{-k}$ for group $k$. The authors further show how to generalize the same technique to linear MDPs.

Overall, the results and techniques in this paper look interesting and thus I recommend acceptance. For improvement, the authors could consider generalizing their results to generalized linear models / kernels / functions with bounded eluder dimension. For example, it seems possible to use the same technique and the algorithm in [1, 2] to achieve uniform-PAC guarantee for functions with bounded eluder dimension. Moreover, is that possible to give a hard instance so that the algorithm [3] fails to achieve uniform-PAC guarantee provably? Currently the authors only show why the analysis [3] fails to provide uniform-PAC guarantee. This still leaves the possibility that the algorithm could achieve uniform-PAC guarantee through a different analysis.

Minor Comment: consider changing $(\epsilon,\delta)$-uniform-PAC in Line 160 to uniform-PAC, since uniform-PAC guarantee is not defined for any specific $\epsilon$ and $\delta$.

[1] Eluder Dimension and the Sample Complexity of Optimistic Exploration

[2] Provably Efficient Reinforcement Learning with General Value Function Approximation

[3] Improved algorithms for linear stochastic bandits

**Time Spent Reviewing:**

5

---

> ### Author Response · Authors · 2021-08-09
> **Response to review Reviewer FX5f**
>
> Thank you for your positive feedback.
>
> Q1: Minor Comment: consider changing $(\epsilon,\delta)$-uniform-PAC in Line 160 to uniform-PAC
>
> A1: Thanks for your suggestion. We will change it in the revision.
>
> Q2:  Achieve uniform-PAC guarantee for generalized linear model/kernel/general function approximation with bounded eluder dimension
>
> A2: Thanks for your insightful suggestion. This is definitely an important future work direction, and we will comment on it in the final version. Some of these extensions (generalized linear models and kernels) are easy, but others (general function approximation with bounded eluder dimension) need much more effort, since in general function approximation, we need a new set of tools to bound the sub-optimality of value functions. We will leave it for future work.
>
>
>
> Q3: Is it possible to give a Hard instance that OFUL fails to be uniform-PAC?
>
> A3: We think so. Here we present a hard instance for which a variant of the OFUL algorithm cannot be uniform-PAC. In the original OFUL algorithm [1], following their notation, the agent selects the action by $x_k = \arg \max_{(x,\theta) \in D_{k}\times \Theta_{k-1}} \langle x, \theta \rangle$. Here we consider a variant of OFUL, where the agent selects the action by $x_k = \arg \max_{(x,\theta) \in D_{k}\times \Theta_{k-1}\cap B(1)} \langle x, \theta \rangle$, where $B(1)$ is a unit ball centered at zero. We consider a special contextual linear bandit problem with dimension d=2, $\theta^*=(0,1)$, and zero noise. The action set in the first K (K is an arbitrary parameter that can be chosen later) rounds is $\{(1,0),(-1,0)\}$ and the action set in the following $\log K$ rounds is $\{(0,1),(0,-1)\}$. So the reward in each step can only be 1 or -1. The agent will randomly choose one action if both actions attain $\arg \max_{(x,\theta) \in D_{k}\times \Theta_{k-1}\cap B(1)} \langle x, \theta \rangle$. We can show that, in the first $K$ round, the confidence radius increases since the determinant of the covariance matrix increases, and it will not provide any information about the second dimension of the vector $\theta^*$ since the two actions are orthogonal to $\theta^*=(0,1)$. After the first $K$ rounds, the confidence radius will be in the order of $\log K$,
> and the covariance matrix $V_K$ is a diagonal matrix and in the order of $diagonal (K,\log K)$. We can show that both $\theta = (0,1)$ and $\theta = (0,-1)$ belong to $\Theta_{k-1}\cap B(1)$, and thus attain the maximum of $\arg \max_{(x,\theta) \in D_{k}\times \Theta_{k-1}\cap B(1)} \langle x, \theta \rangle$. Therefore, the agent will almost ‘randomly’ pick one of the two actions in the later $\log K$ rounds. The random selection leads to a 1-suboptimality gap for about half of the $\log K$ rounds, which indicates that OFUL cannot be uniform-PAC for any finite $f(\epsilon, \delta)$ on this bandit problem, by selecting $\log K > f(\epsilon, \delta)$.
>
> For the original OFUL, we believe a similar hard instance can be constructed with some extra effort. We will add a discussion on it in the final version.

---

### Decision · Program_Chairs · 2021-09-27

**Decision:**

Accept (Poster)

**Comment:**

The reviewers find this to be a theoretically strong paper.  I see no reason to disagree and acceptance seems warranted.  However, I feel that the technical notion of a uniform PAC bound is mainly of theoretical interest.  It not clear how much practical insight uniform bounds provide over the insights already provided by standard PAC bounds.  A poster seems appropriate.